# A star-nose-like tactile-olfactory bionic sensing array for robust object recognition in non-visual environments

Mengwei Liu[1,2,12], Yujia Zhang[1,2,12], Jiachuang Wang[1,2,12], Nan Qin[1], Heng Yang[1,2], Ke Sun[1,2], Jie Hao[3], Lin Shu[3], Jiarui Liu[3], Qiang Chen[4], Pingping Zhang[5] & Tiger H. Tao [1,2,6,7,8,9,10,11✉]

Object recognition is among the basic survival skills of human beings and other animals. To date, artificial intelligence (AI) assisted high-performance object recognition is primarily visual-based, empowered by the rapid development of sensing and computational capabilities. Here, we report a tactile-olfactory sensing array, which was inspired by the natural sense-fusion system of star-nose mole, and can permit real-time acquisition of the local topography, stiffness, and odor of a variety of objects without visual input. The tactile-olfactory information is processed by a bioinspired olfactory-tactile associated machine-learning algorithm, essentially mimicking the biological fusion procedures in the neural system of the star-nose mole. Aiming to achieve human identification during rescue missions in challenging environments such as dark or buried scenarios, our tactile-olfactory intelligent sensing system could classify 11 typical objects with an accuracy of 96.9% in a simulated rescue scenario at a fire department test site. The tactile-olfactory bionic sensing system required no visual input and showed superior tolerance to environmental interference, highlighting its great potential for robust object recognition in difficult environments where other methods fall short.

[1] State Key Laboratory of Transducer Technology, Shanghai Institute of Microsystem and Information Technology, Chinese Academy of Sciences, Shanghai 200050, China. [2] School of Graduate Study, University of Chinese Academy of Sciences, Beijing 100049, China. [3] Institute of Automation, Chinese Academy of Sciences, Beijing 100049, China. [4] Shanghai Fire Research Institute of MEM, Shanghai 200003, China. [5] Suzhou Huiwen Nanotechnology Co., Ltd, Suzhou 215004, China. [6] Center of Materials Science and Optoelectronics Engineering, University of Chinese Academy of Sciences, Beijing 100049, China. [7] 2020 X-Lab, Shanghai Institute of Microsystem and Information Technology, Chinese Academy of Sciences, Shanghai 200050, China. [8] School of Physical Science and Technology, ShanghaiTech University, Shanghai 200031, China. [9] Institute of Brain-Intelligence Technology, Zhangjiang Laboratory, Shanghai 200031, China. [10] Shanghai Research Center for Brain Science and Brain-Inspired Intelligence, Shanghai 200031, China. [11] Center for Excellence in Brain Science and Intelligence Technology, Chinese Academy of Sciences, Shanghai 200031, China. [12] These authors contributed equally: Mengwei Liu, Yujia Zhang, Jiachuang Wang. ✉email: tiger@mail.sim.ac.cn

The ability to effectively observe different objects and to accurately recognize targets is a perceptive skill that animals have developed during their evolutionary history. Robust object recognition in sophisticated environments for automobiles and robots is a research topic that has raised a lot of interest in the scientific community over recent years[1–3]. Many approaches have been proposed and most of them are visual-based[4–11]. However, interferences such as unclear objects (occlusions) or poor light conditions can severely impact the accuracy when performing visual object recognition[5,12]. Recently, the combination of visual information and other sensing modalities, such as somatosensory and auditory sensing, has achieved notable progress[3,5,13–16].

In addition to vision, tactile and olfactory perceptions are two other crucial natural capabilities that animals have developed in order to achieve object recognition[17–19]. For example, star-nosed moles have evolved the capability of object recognition using only tactile and olfactory perception thanks to the nerve-rich appendages around its nose, allowing it to survive in the lightless underground environment[20–22]. In fact, the mole's visual-related nervous area is replaced with the tactile perceptive and fusion area during the embryonic period as a naturally evolved trade-off[23]. This type of biological strategy demonstrates the advantages of tactile-olfactory fusion in object recognition, including compact sensory constituents, high accuracy, excellent environmental suitability, high efficiency, and low power consumption.

In this work, we report a star-nose-like tactile-olfactory sensing array mounted on a mechanical hand which permits the real-time acquisition of an object's local topography, stiffness, and odor when touching the object. The information is gathered and then processed by a bioinspired olfactory-tactile (BOT) associated machine-learning strategy, essentially mimicking the biological fusion procedures in the neural system of the mole. We aim to use the tactile-olfactory intelligent sensing arrays to achieve human identification and support human rescuing in hazardous environments as a proof-of-concept.

## Results

### Bioinspired design of tactile-olfactory sensory system.
Star-nosed moles are considered to have one of the best tactile senses among mammals. The unique nose structure of the star-nosed mole significantly improves its ability for object perception during exploring and foraging[23]. Benefitting from the naturally evolved tactile sensing organs (Eimer's organs) on the 22 epidermal appendages around the nostrils, the star-nosed mole combines the senses of touch and smell together perfectly for rapid detection and predation in dark underground environments with little contribution from vision or audition[22].

The tactile-olfactory sensing and fusion procedures of the star-nosed mole are achieved through the compact linkage between the perceptive organ and the cerebral nervous system. These procedures include the conducting and processing of information from the initial organs to the primary areas (PA), feature extraction and early interactions of the original signals in the association area (AA), and the subsequent multisensory fusion process (Fig. 1a)[24–27].

The general layout of the biological perceptual organs in star-nosed moles has been retained in our design. Figure 1b shows an ancillary mechanical hand that is composed of 5 tactile sensing arrays with 14 force sensors evenly attached on each fingertip (70 force sensors in total) and 1 olfactory sensing array with 6 different gas sensors attached on the palm, allowing effective acquisition of both tactile and olfactory information. Herein, this bioinspired intelligent perception system was designed and applied to achieve the robust object recognition for human

rescue in challenging environments, such as where victims may be buried or where there is the presence of harmful gas. In this work, 11 objects in five categories (i.e., Human, Olfactory interference objects, Tactile interference objects, Soft objects, and Rigid objects) were selected as proof-of-principle recognition targets (Fig. 1c). Among these, the human is the main target to be identified in dangerous situations. In addition to soft and rigid commodities, objects with similar stiffness and odor to those of humans, such as animals and worn clothes were selected as the interference objects in order to test the recognition system because they are usually difficult to distinguish from humans solely by single modal sensing.

To mimic the rapid decision-making of the mole, our BOT associated architecture consists of three neural networks resembling the olfactory and tactile signal fusion hierarchy in the mole brain (Fig. 1d). First, a convolutional neural network (CNN) and a fully connected network were used for early tactile and olfactory information processing, which resembles the function of the local receptive field of biological nervous systems and thus mimics the initial processing of tactile and olfactory information in the PA. Second, two fully connected networks were used for extracting the features from the original information and for making pre-decisions about the output weights of the tactile and olfactory information based on the surrounding environment at the same time, mimicking the early interactions of the original signals in the AA of biological nervous systems. Afterwards, three more fully connected networks are used for multisensory fusion, resembling the biological information fusion process.

### Design and functionality of individual sensors and integrated sensing arrays.
We have designed and fabricated a series of silicon-based force and gas sensors with high sensitivity and stability. The sensors were transferred and integrated in arrays on flexible printed circuits for compliant attachment onto the ancillary mechanical hand (Fig. 2a). The small footprint ($0.5 \times 0.5$ mm$^2$) of the force sensors ensured the high resolution of our tactile sensing for object recognition[28,29]. Appropriate protections have been implemented to improve device robustness under harsh conditions (Supplementary Fig. 1)[30–32]. Furthermore, six gas sensors ($3 \times 3$ mm$^2$ for a single unit) compose one olfactory sensing array, and each sensor is functionalized to be highly sensitive to one particular gas, resulting in a customizable capability for specific perception of the odor of the detected object in a complex environment, mimicking biological olfactory receptor cells[33–39] (see "Methods").

High sensitivities provided accurate sampling data for the following analysis (Fig. 2b)[28–30,34–41] (Supplementary Figs. 2 and 3 and Supplementary Notes 1 and 4). In this work, these force sensors could accurately reflect the multiple statuses during the contact procedure in a typical object interaction sequence with a sensitivity of 0.375 mv kPa$^{-1}$ over a range of 0−400 kPa (Fig. 2c). Figure 2d indicates that the force sensor performs well in the identification of objects with various elastic stiffness (Supplementary Fig. 4 and Supplementary Notes 2 and 3). The gas sensors demonstrated rapid responses during contact with the detected gas flow (response phase of ethanol gas sensor for example, Fig. 2e), which took approximately ten seconds before reaching the steady phase. As shown in Supplementary Fig. 5, the performance of our gas sensor keeps steady over 60 days, providing an accurate olfactory dataset for object recognition. The sensor recovered to the initial status within ten seconds after the gas was cut off. Gases with different concentrations and types could be distinguished (Supplementary Fig. 6). Both force and gas sensors showed high stability from −20 to 60°C, providing robust performance under challenging conditions (Supplementary Fig. 7).

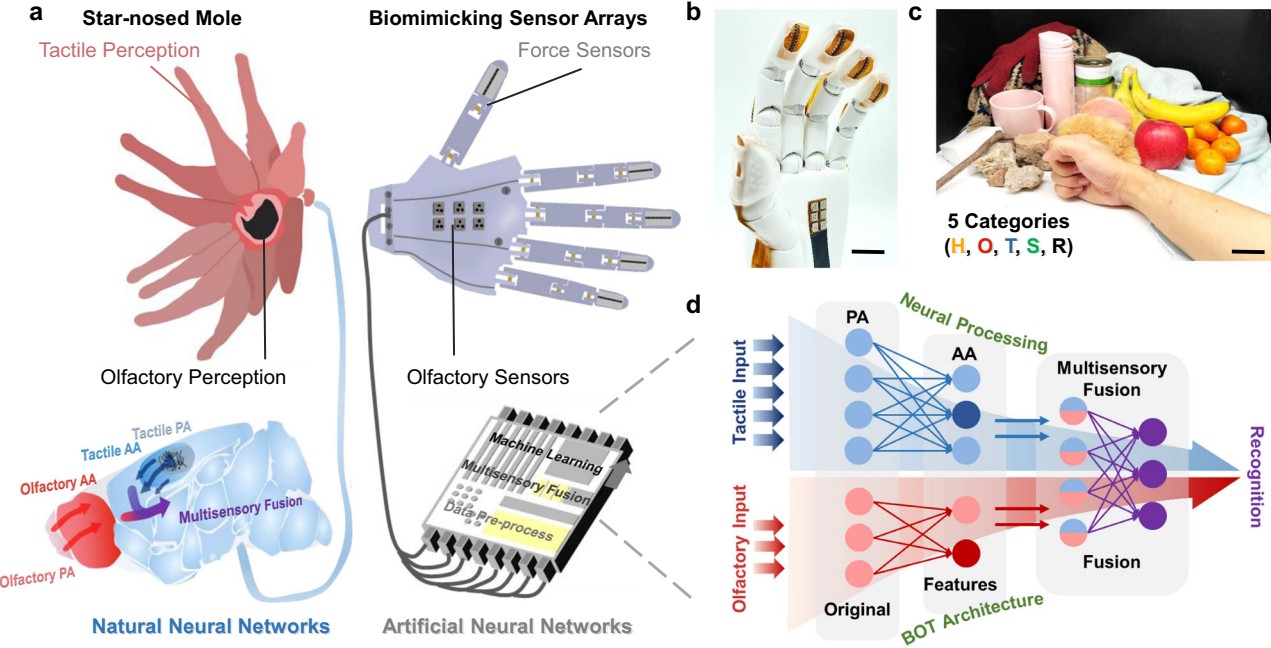

**Fig. 1 Bioinspired tactile-olfactory associated intelligent sensory system. a** Schematic illustration of the bio-sensory perceptual system in the star-nosed mole (left) and biomimicking intelligent sensory system (right). Top left: diagram of the unique structure of the star-shaped nose. Bottom left: Scheme showing the processing hierarchy of tactile and olfactory information in the neural system of the star-nosed mole. Blue and red areas represent the processing region (PA, primary area; AA, association area) for tactile and olfactory information, respectively. Blue and red arrows represent the direction of tactile and olfactory information flow; the purple arrow shows the information flow of the multisensory fusion. Top right: schematic diagram of the sensing array on the mechanical hand, including force and olfactory sensors. Bottom right: illustration of artificial neural networks. Images of the mechanical hand with scale bar of 2 cm (**b**) and eleven different objects to be identified, scale bar: 3 cm (**c**). Objects to be tested can be divided into five categories including Human (**H**, main target), Olfactory interference (**O**, e.g., worn clothes), Tactile interference (**T**, e.g., mouse), Soft objects (**S**, e.g., fruits), and Rigid objects (**R**, e.g., debris). **d** The machine learning framework consists of three connected layers of neural networks that mimic the multisensory fusion process hierarchy. Top left: early tactile information processing. Bottom left: early olfactory information processing. Right: neural network resembling the high-level fusion of tactile and olfactory interactions.

We next performed a demonstration of human arm recognition to test the performance of the sensing arrays (Fig. 2f). It is worth noting that we focused on the tactile perception over the fingertips, rather than identifying objects based on their overall shape[17]. Therefore, we could prominently reduce the spatial mapping pixels and corresponding data complexity, while maintaining the high accuracy via perceiving object stiffness and local topography (See details in Supplementary Figs. 2, 8, 9 and Notes 2 and 3). Figure 2g presents the gradually increasing output voltage at three sequential feature points (i.e., the minimum force, maximum gradient, and maximum force), consistent to Fig. 2c. Tactile mappings captured the key features of local topography and the material stiffness of the objects in real-time, while the olfactory array presented excellent recognition capability for distinguishing a human arm from other objects (Fig. 2h). In addition, our tactile and olfactory sensing arrays also present the capability of detecting objects covered with water or mud, which is common in real rescue scenarios. As shown in Supplementary Fig. 10, such interferences of olfactory perception can be alleviated after combining olfactory sensing with tactile perception, maintaining the high recognition accuracy (See details in Supplementary Notes 4 and 5).

To implement a recognition task based on BOT-associated learning, we built a custom tactile-olfactory dataset containing 55,000 samples distributed into 11 types of objects covering five categories. Each sample consists of one group of output voltage data captured from 70 force sensors and a group of output resistance data from six gas sensors. We used the t-distributed stochastic neighbor embedding, a dimensionality reduction

technique, in order to visualize the tactile and olfactory data (Supplementary Fig. 11)[42]. Each point on the tactile/olfactory data plot represents the corresponding sensory information of one object projected from the 70D/6D data into two dimensions. The points of the same object type were clustered together, forming 11 categories of objects. Grasping gestures can cause a difference in the tactile array response and therefore result in multiple clusters for one object in the plot, which can be distinguished in our system.

**Design and recognition performance of the BOT algorithm.** Figure 3a shows the framework of the BOT associated learning architecture, including a versatile CNN for extracting tactile information from time-variant tactile mappings[17], a single fully connected neural network for obtaining olfactory information, and a three-layer fully connected neural network (with a 0.5 dropout rate) for final associated learning. The sparse connectivity of the neural network enhanced the generalization ability of the BOT architecture for scalable sensory data fusion considering the different data formats of tactile and olfactory signals (dimensionality, temporary density, and sparseness). In the fusion procedure, a scenario-dependent feedback was added in the BOT associated learning network, which enabled a tunable weight ratio between tactile and olfactory information. According to actual applications, when one input perception is severely disturbed or damaged by the environment, the BOT network could be adjusted to rely on the other one by increasing the relative weight of another perception, ensuring a high recognition rate for objects in challenging scenarios.

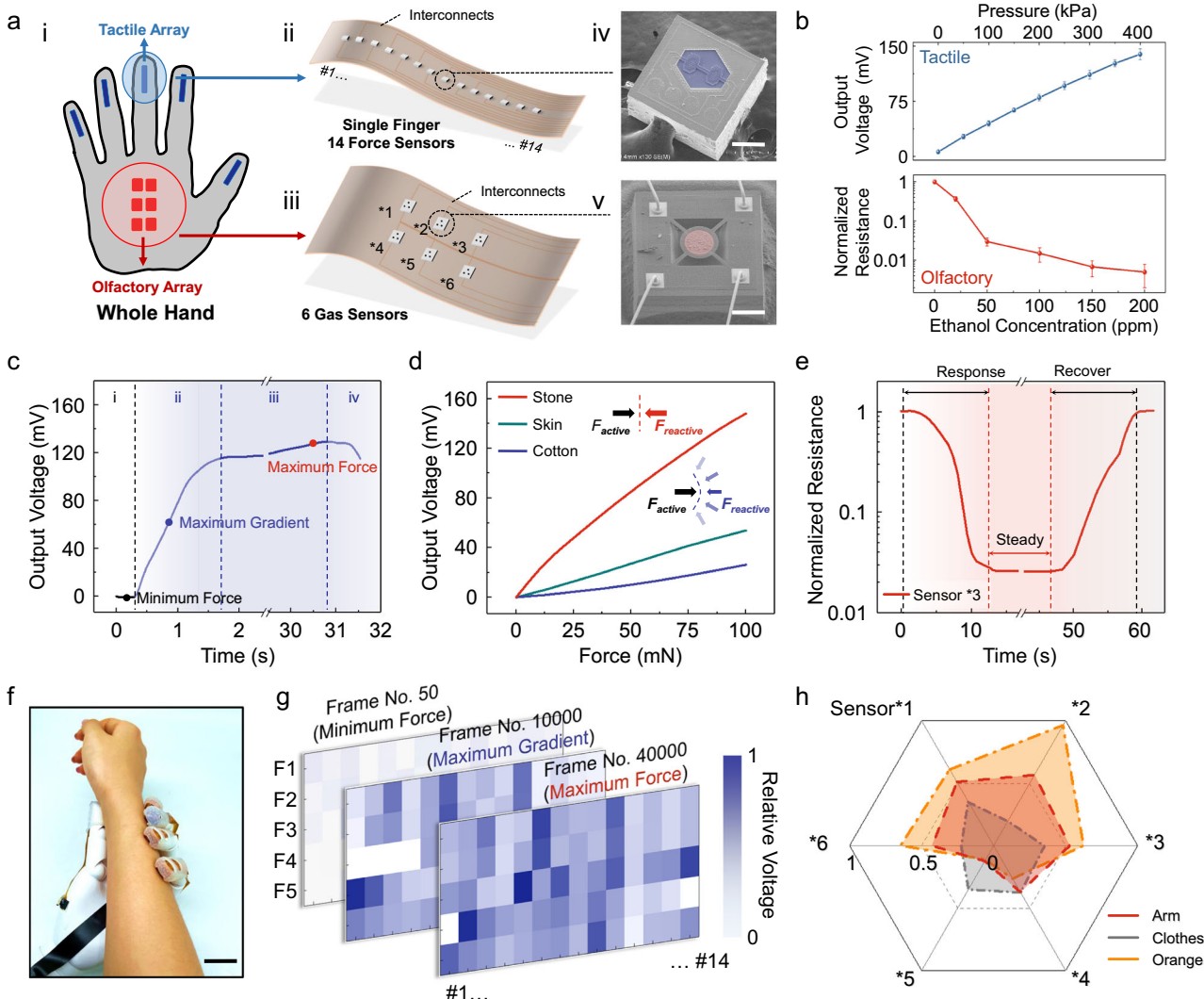

**Fig. 2 Characterization of tactile and olfactory sensing array. a** The design of the array architecture (i) shows the location of the 14 force sensors on each fingertip (ii), along with the location of the six gas sensors on the palm (iii). The Si-based force (iv) and gas (v) sensors are fabricated using microelectromechanical systems techniques and integrated on flexible printed circuits. Blue area: force-sensitive area (single-crystalline silicon beam) with scale bar of 200 μm. Red area: area modified by gas-sensitive material with scale bar of 400 μm. **b** The sensitivity of the force and gas sensors. $n = 12$ for each group. The error bars denote standard deviations of the mean. Top: the output voltage response of the force sensor under gradient pressure loading. Bottom: The normalized resistance response of the gas sensor in the continuously increasing ethanol gas concentration. **c** In a typical touching process, the fingers increasingly get closer to the object until the point of contact (i. reach phase) and experiences a sudden rise in tactile forces as the object is touched (ii. load phase). Then the hand would hold the object for a certain time (iii. hold phase) and at last release the object (iv. release phase). **d** Response curve of the force sensor during the process of contacting three different objects. When the same force is applied to the objects by the mechanical hand, the deformation degree of the objects varies according to their different elasticity modulus, changing the local contact area and consequent reactive pressure. **e** The normalized resistance response curve of the gas sensor during the process of contact and separation from the detected gas flow. **f** Photograph of the mechanical hand touching a human arm, scale bar: 2 cm. **g** The tactile mappings at three different feature points in (**c**). Each contains 70 pixels with 14 pixels for each fingertip. **h** The hexagonal olfactory mappings of three different objects including an arm, worn clothes, and an orange.

Moreover, we also implemented two unisensory learning approaches for object recognition, including sole tactile-based recognition using only tactile data based on a CNN, and sole olfactory-based recognition using only olfactory data based on a feedforward neural network (Supplementary Fig. 12). The confusion matrixes for these approaches showed that, in a testing dataset containing 11,000 samples, BOT associated learning has a higher accuracy for correct recognition (91.2%) than tactile-only recognition (81.9%) and olfactory-only recognition (66.7%) (Fig. 3b, c and Supplementary Fig. 14a, b), proving the importance of multi-modal sensing and fusion.

We further optimized the BOT associated learning by altering the eigenvalue extraction method of the original data and the data output mode for objects recognition using the same training and testing tactile-olfactory dataset, implementing fusion based on random points extraction (BOT-R), fusion based on feature points selection (BOT-F), and fusion based on feature points selection and multiple data output mode fusion (BOT-M) (Supplementary Fig. 13). The accuracy of object recognition was improved significantly following learning optimization with the highest recognition rate (96.9%) for BOT-M (Fig. 3d). The recognition accuracy of the optimized learning architecture began

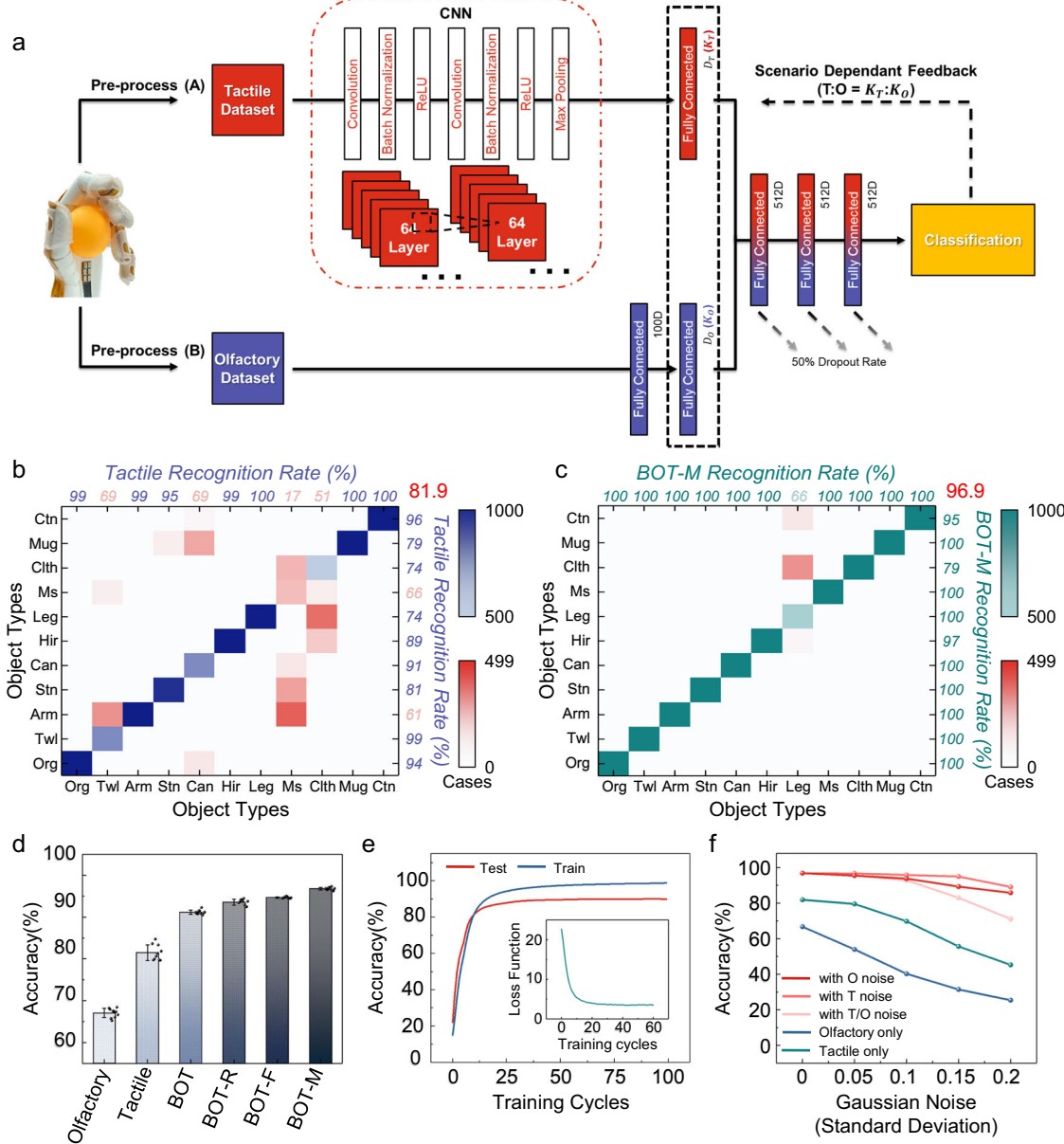

**Fig. 3 BOT associated learning for object classification. a** Scheme showing how tactile and olfactory information is processed and fused in the BOT associated learning architecture. 512D, 512-dimensional vector; 100D, 100-dimensional vector. **b** Confusion matrix of the sole tactile recognition strategy. **c** Confusion matrix of the BOT-M recognition strategy. The full name of the abbreviation: Org-Orange; Twl-Towel; Arm-Arm; Stn-Stone; Can-Can; Hir-Hair; Leg-Leg; Ms-Mouse; Clth-Worn Clothes; Mug-Mug; Ctn-Carton. **d** BOT-M associated learning shows the best accuracy among the unimodal (tactile and olfactory) and multimodal fusion strategies (BOT, BOT-R, BOT-F, BOT-M). n = 10 for each group. The error bars denote standard deviations of the mean. Unimodal strategies: olfactory-based recognition using only olfactory data and tactile-based recognition using only tactile data. Multimodal fusion strategies using both tactile and olfactory data: BOT associated learning fusion, fusion based on random points (BOT-R), fusion based on feature point selection (BOT-F), fusion based on feature point selection and multiplication (BOT-M). The final recognition accuracies are 66.7, 81.9, 91.2, 93.8, 94.7, and 96.9% for olfactory, tactile, BOT, BOT-R, BOT-F, and BOT-M strategies, respectively. **e** The change in recognition accuracy of the BOT-M neural network with the increase of training cycles. Inset: Loss of function variation during the training process. **f** The testing results of tactile-, olfactory-, and BOT-M-based strategies under defective tactile and olfactory information with various Gaussian noises (0.05, 0.1, 0.15, and 0.2) show that BOT-M can maintain higher recognition accuracy with increased noise levels compared to unimodal fusion strategies.

convergent and remained stable after around 20 training cycles for both testing and training datasets (Fig. 3e). We further evaluated the influence of both tactile and olfactory noise on the recognition accuracies of these trained models (olfactory, tactile, and BOT-M-based recognition strategies) by adding Gaussian white noise in the testing dataset. The increased noise level significantly deteriorates the recognition accuracies of the unisensory strategies, while BOT-M continues to maintain high recognition accuracies (Fig. 3f).

**Human recognition in challenging conditions**. We adapted our sensing system for human recognition as demonstrated in a simulated rescue scenario at a fire department test site. Figure 4a depicts the system, consisting of a mechanical hand equipped with tactile-olfactory sensing arrays, a data pre-processing unit for capturing the tactile and olfactory information from the sensing array, and a data-fusion unit for implementing the BOT-M associated learning and final object recognition. Four different scenarios with various obstructions were built in terms of (1) gas

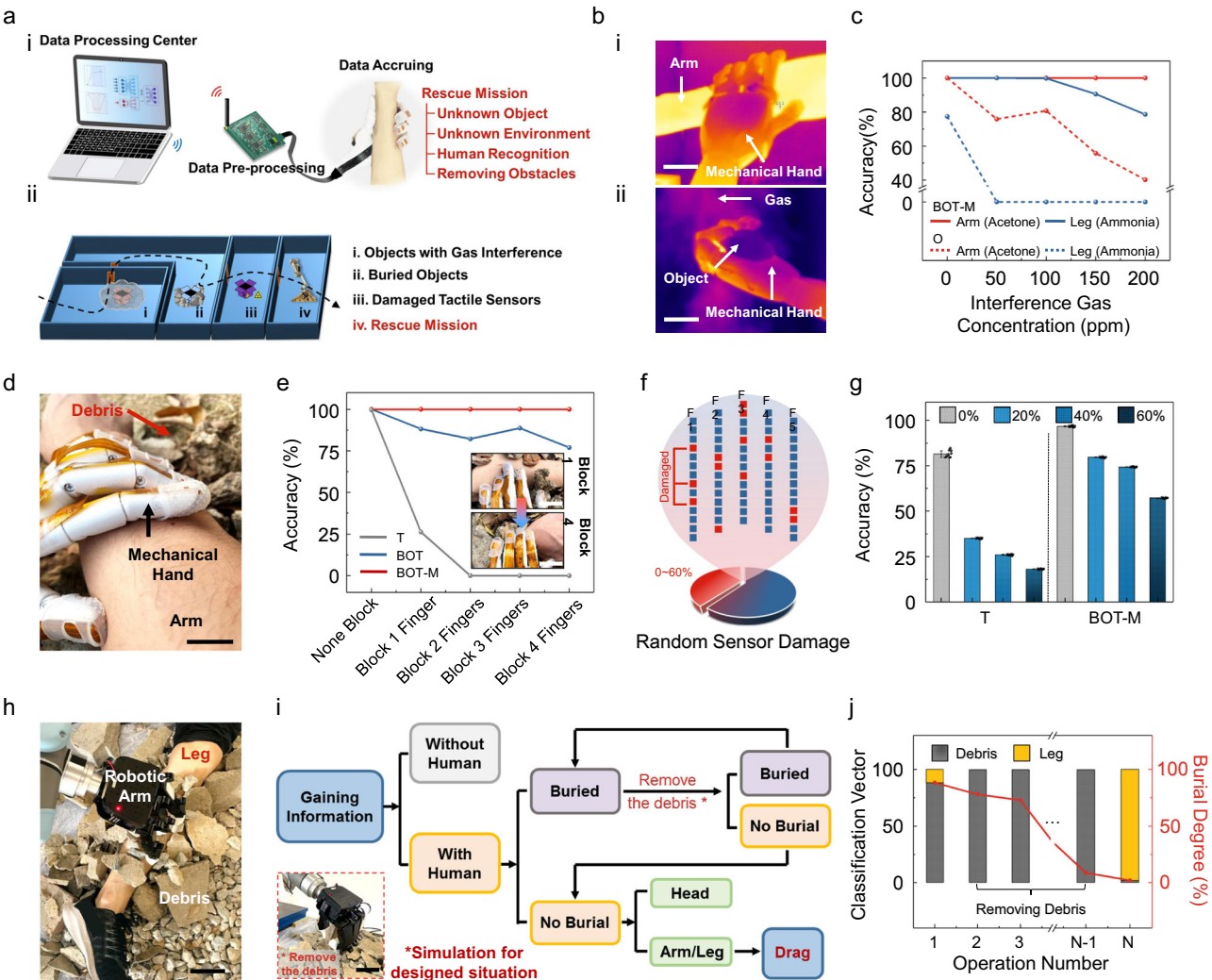

**Fig. 4 Human recognition in a hazardous environment based on BOT. a** Schematic illustration of the testing system and scenarios. (i) Scheme showing the system consisting of a computer, a wireless data transmission module, a data pre-processing circuit board, and a mechanical hand. (ii) Four different hazardous application scenarios including gas interference, buried objects, partially damaged tactile sensors, and simulated rescue mission. **b** IR photographs of the mechanical hand holding different objects (i. an arm and ii. other objects) with various gas interference, scale bar: 4 cm. **c** Recognition accuracy for different parts of the human body under the interference of various gas concentrations (50, 100, 150, and 200 ppm) of acetone and ammonia using olfactory-based recognition and BOT-M associated learning. **d** Photograph of arm recognition underneath the debris, scale bar: 1.5 cm. **e** The change of arm recognition accuracy as burial level continues to increase using tactile-based recognition, BOT, and BOT-M associated learning. Inset: Photos of one finger (Top) and four fingers (Bottom) of the mechanical hand being blocked from touching the arm, scale bar: 5 cm. **f** Scheme showing damage to random parts of both force and gas sensors in the tactile array. **g** The accuracy for arm recognition with different sensor damage rates using tactile-based recognition and BOT-M associated learning. $n = 10$ for each group. The error bars denote standard deviations of the mean. **h** Photograph of a volunteer's leg buried underneath debris and a robotic arm performing the rescue mission, scale bar: 8 cm. **i** Flow diagram shows the decision-making strategy for human recognition and rescue. Inset: Photograph of the robotic arm removing the debris, scale bar: 10 cm. **j** The variation of leg/debris classification vector and the alleviated burial degree while the covering debris being gradationally removed by the robotic arm.

interferences, (2) buried objects, (3) partially damaged tactile sensors, and (4) simulated rescue mission.

We first tested the recognition performance of different body parts, such as the arm and leg, using the multisensory fusion system in environments with different inference gases of various concentrations. In order to simulate the practical scene and to prove the system's capability of resisting disturbance, acetone and ammonia were chosen to interfere with the recognition of the human body. Specifically, acetone simulates the emergent situation of gas leakage in an industrial accident, while ammonia possesses a similar odor to that of the human body (Fig. 4b). The recognition accuracies for both unisensory and multisensory approaches (olfactory and BOT-M) under acetone and ammonia

interferences are shown in Fig. 4c. When the concentration of acetone increased, the accuracy of the sole olfactory approach decreased dramatically, whereas the tactile-olfactory fusion method maintained high accuracy (>99%), showing excellent human recognition performance in the situation of a dangerous gas leak. Similarly, with the increase of ammonia concentration, the multisensory fusion approach maintained high accuracy (>80%) of human recognition under the disturbance of similar odor when compared to the rapid decline of accuracy for sole olfactory recognition. Meanwhile, the system can also sense the presence of the inferencing gases (Supplementary Fig. 15), which could provide warning for timely evacuation if needed. The robust performance of the BOT system under gas disturbance

primarily depends on tactile compensation while olfactory perception is disturbed.

In a burial scenario that involves visual obstruction, tactile-olfactory sensing plays a major role in object and environment perception. However, for sole tactile perception of the object's overall shape, the obstruction of debris will cause serious deviation, resulting in a decline of recognition accuracy[43]. In contrast, the local topography and material stiffness perception of our tactile arrays could achieve higher accuracy when the target is partially exposed, such as partially buried human body (Fig. 4d). As proof-of-concept, one to four fingers of the mechanical hand are blocked in order to simulate different degrees of burying (Supplementary Fig. 16). The results in Fig. 4e show that when the burial level increases, the accuracy of the sole tactile perception decreases dramatically, whereas the tactile-olfactory fusion method maintains its initial high accuracy (>99%). One accountable reason is that the force sensor can distinguish objects with various stiffness, thus distinguishing the soft human arm from the rigid debris with high accuracy of tactile perception could be achieved when there is no block for the mechanical hand. In addition, by increasing the weight proportion of olfactory information, identification of human odor can be a strong supplement to the lack of tactile perception, leading to an improvement in the accuracy of human body recognition in burial scenarios with tactile olfactory fusion method.

Importantly, it is possible that the tactile and olfactory sensing array could endure partial failure due to contact with sharp objects or damaging radiation during practical application. Thus, we also intentionally disabled some of the sensors randomly, with a failure rate varying from 0 to 60% (Fig. 4f). In this case, the tactile-olfactory fusion method displays excellent recognition accuracy compared to the sole tactile perception (Fig. 4g). On one hand, the result is attributed to the complementary effect of the olfactory information to supplement the defected tactile information in the multisensory fusion process. On the other hand, the scenario-dependent feedback in the BOT associated learning can accommodate harsh environments and can improve accuracy via changing the tactile and olfactory weight (see "Methods").

On top of the search and discovery of human existence in these challenging conditions, the rescue mission may also include removing obstacles and evaluating the burial degree. As proof-of-concept, we combined our sensing system with a dexterous robotic arm for the demonstration of removing the debris and then rescuing the buried human (Fig. 4h). The design of the robotic arm focuses more on handling debris so is different from the previous mechanical hand used for object recognition. In detail, the degree of burial is determined by the relative proportion of classification vectors of debris and the exposed human body in the detected area. Following the rescue procedures shown in Fig. 4i, our sensing and BOT system could evaluate the existence and burial degree of human if needed, guiding the robotic arm to automatically remove the covering debris until the buried body part being fully exposed. As shown in the Supplementary Movies 1 and 2, we mounted the mechanical hand on a robotic arm to recognize and grab the debris, and then move them away from the designated area; then we let the same robotic system to touch and recognize the fully-exposed human arm/leg. Figure 4j shows such process of the step-by-step debris removal and corresponding reduced burial degree. In the end, the fully exposed body part can be recognized by our system and could furnish a rescue window for dragging and healthcare.

## Discussion

Odor is the chemical fingerprint of every object, but until now has rarely been combined with tactile sensing for object identification. Many recognition modalities already published do not use the explicit combination of tactile and olfactory sensing[44,45], mainly because gas sensors have both dimensional and temporal data mismatch with force sensors, and are susceptible to ambient gas interference. Nevertheless, by the effective data preprocessing and complementary setting of 6 different gas sensors, we have proved that olfactory sensing can also be suitable for feature fusion with tactile data. Therefore, the olfactory sensing provides an alternative option for object recognition, which essentially carries more information of objects compared to other ordinary physical parameters, such as temperature or humidity.

Comparing with the vision-based sensing systems[2], our tactile and olfactory fusion strategy has a relatively small input data size, leading to smaller requirements of computing resources and faster identifications, which are crucial in the rescue mission. In addition, for recent studies about objects identification using tactile sensors, most of them choose to use flexible none-silicon-based strain sensors[17]. However, in our design, we use silicon-based force/gas sensors fabricated by micro-electromechanical systems technologies, which have more robust performance, smaller size, and higher accuracy. Meanwhile, our strategy for the first time takes odor as one input modality and thus is more suitable for situations that gas plays an important role, such as human recognition in rescue scenarios.

We have reported a star-nose-mimicking tactile-olfactory sensing system combined with machine learning architecture to achieve robust object recognition under challenging conditions. Using silicon-based force and gas sensors with high sensitivity and stability, the flexible sensing arrays on the mechanical hand could acquire reliable tactile-olfactory information by touching the object. We have developed a BOT-associated machine-learning strategy to extract key features about the local topography, material stiffness, and odor of the tested object. By fusing tactile and olfactory information together, our BOT-based architecture could classify objects against environmental interferences with an accuracy of 96.9% and offer excellent human identification performance (accuracy > 80%) under the hazardous scenarios of gas interference, object burying, damaged sensors, and rescue mission. Compared to visual perception, our tactile-olfactory sensing strategy orchestrated an alternative approach in dark or blocked spaces and exhibited its superiority for human identification in rescue conditions.

## Methods

**Fabrication of the force sensor.** After thermal oxidization on the front side of the silicon wafer, the first-time photolithographic steps were performed to define the locations of the piezoresistors. The piezoresistors were formed by boron ion-implantation followed by the drive-in process. Then a 0.3 μm thick low-stress silicon nitride layer and a tetraethoxysilane layer with a thickness of 0.8 μm were formed by low-pressure chemical vapor deposition (LPCVD). Then the second photolithography was conducted to pattern the cavity-releasing micro-holes with silicon deep reactive ion etching (RIE). Then, the low-stress silicon nitride film with a thickness of 0.2 μm and tetraethoxysilane film with a thickness of 0.2 μm were sequentially deposited by LPCVD. RIE was used to selectively etch off the deposited low stress silicon nitride and tetraethoxysilane composite layer at the trench bottom to expose bare silicon at the bottom surface of the holes. After that, silicon deep RIE was processed again to deepen the holes, forming pressure reference cavity. Then 40% aqueous KOH was used to complete the inter hole cavity-release by lateral under etch. Subsequently, LPCVD was used again to form a layer of conformal poly-silicon with a thickness of 4 μm for the seal of the sensor. After that, a deeper trench-etch was processed by deep RIE to define the shape of the cantilever structure and the structure was then released into free-standing with wet etching by aqueous KOH (25%). Finally, the interconnection lines of the piezo-resistive Wheatstone bridge were formed by sputtering a layer of Al film with a thickness of 0.1 μm. Further details are in Supplementary Note 1. Please also check previously reported works for the basic version of fabrication procedures and detailed characterizations[28,30].

**Fabrication of the gas sensor.** In the convenience of silicon anisotropic wet etching, we used the silicon wafer with (100) surface as the device substrate. Silicon

oxide and silicon nitride multilayer composite membrane were selected as devices support layer for better insulation. In detail, a combination of dry and wet oxygen thermal oxidation method was used to fabricate 200 nm silicon oxide and LPCVD was used to make silicon nitride with a thickness of 1000 nm. The Ta/Pt heating resistance wire and pad were fabricated by lift-off process, with a thickness of 300 and 3000 Å. The composite film of silicon oxide (2000 Å) and silicon nitride (4000 Å) deposited by PECVD was used as the isolation layer. Then the RIE process was used to etch the isolation layer to expose the heating electrode below. The pair of Ta/Pt cross finger electrodes and pad were fabricated by lift-off process with a thickness of 300 and 3000 Å. After that, the exposed silicon oxide and silicon nitride composite films were etched thoroughly by RIE process, and the substrate silicon was exposed to form a window for the following film releasing. At last, the structure was released by anisotropic wet etching in tetramethylammonium hydroxide solution for 4 h and then gas-sensitive material was modified.

The sensing materials of these six gas sensors are: (1) carbon nanotubes modified by magnesium oxide particles; (2) carbon nanotubes modified with platinum particles; (3) graphene modified with copper oxide particles; (4) platinum-doped tin oxide; (5) platinum-doped tungsten oxide; (6) composite material of zinc oxide and tin oxide.

Respectively, these six gas sensors were sensitive to six different gases, including ethanol, acetone, ammonia, carbon monoxide, hydrogen sulfide, and methane. Further details are in Supplementary Note 4. Please also check previously reported works for the basic version of fabrication procedures and detailed characterizations[40,41,46].

**Integration of the force and gas sensors**. We used Altium Designer software for the design of the flexible printed circuit. Force sensors were fixed on the flexible printed circuit with vinyl. In order to protect the force sensor from external damage during practical applications, further protections were implemented to the sensing arrays after wire bonding. First, the silver paste was applied at the wire binding node on the flexible printed circuits in order to strengthen the contact. Then the printed circuits were placed on hot plate for 30 min at 160 °C for solidification process. After that, vinyl was applied on the wire bonding area and then stayed on hot plate for 45 min at 105 °C. For gas sensors, after each sensor was packaged, six sensors were soldered on the flexible printed circuit following similar procedures. At last, silica gel was applied on the surface of the force sensor and curing for 24 h to solidify. After integration, the sensing arrays were attached on a commercial mechanical hand for further test.

**Characterization of the tactile-olfactory sensing arrays**. The sensing arrays were connected to a home-built data serial bus for powering and signal pre-processing. An instrumentation amplifier array (AD 8221) and a data acquisition card (NI 6255) were used to amplify the output voltage signals forty times and collect the amplified signals. A portable resistance detection unit was used to measure the resistive gas responses. Digital force measurement equipment was used to apply external force on the tactile arrays. The sensing arrays were put in an 18 L glass chamber for quantitative gas detection. The gas concentration was controlled by a dynamic gas pumping system. A LabVIEW program was used to gather the data with a sampling frequency of 50,000 points per sec from different channels for the next step. All experiment participants were fully voluntary and the construction of all challenging scenarios are under the guidance from the Shanghai fire department.

**Detailed touching process and corresponding tactile responses**. As shown in Fig. 2c, the mechanical hand gradually approached to the object in the beginning, showing relatively stable proprioceptive signal in the tactile map (reach phase); the gray dot showed the local minimum value. When contact started (load phase), the output voltage of the tactile array increased suddenly, resulting in a steep temporal gradient; the blue dot showed the location of the local maximum gradient. Then in the third phase (hold phase), since the mechanical hand kept contacting with the objects with a fixed posture, the output voltage maintained a certain value with some slight variations; the red dot showed the local maximum value. In the end, as the mechanical hand separated from the target, the output value decreased at the same time (release phase).

**Detailed olfactory responses**. The results presented in Supplementary Fig. 5 detailed the responses of these six gas sensors in the same gaseous ethanol environment, showing different outputs due to various gas-sensitive material modifications. Because of these unique features of our olfactory sensing array, gases with different concentrations and types could be distinguished accurately (Supplementary Fig. 6). Especially, for some similar gases, such as methanol and ethanol, the olfactory sensing array preserved excellent recognition capability on account of the sensors selection and complementation (numbers 2 and 5).

**Dataset preparation and design of the machine-learning architecture**. Eleven detected objects: orange, towel, stone, can, worn clothes, carton, mug, mouse, hair, leg, and arm (Supplementary Fig. 17). During the experiment, Balb/c mice (6–8 weeks old, male, Shanghai SLAC Laboratory Animal Co., Ltd, China) were kept under room temperature and humidity. The mechanical hand, covering with

the tactile-olfactory sensing arrays, was controlled to touch the object and hold it for a minute subsequently. Both tactile and olfactory information were collected and saved in the computer simultaneously for further analysis. To allow for device variation and hysteresis, the raw olfactory data was first normalized at the pre-processing procedure. We build the machine-learning architecture using the PyTorch deep learning framework. Further details are in Supplementary Notes 5 and 6.

Spurring by the rapid development of machine learning techniques, especially the widely used deep convolutional neural networks (CNNs), object recognition could be divided into several steps, including image capturing, categories labeling, data training, and target identification based on probability distribution. For the two unisensory learning approaches for objects recognition, the training and testing samples were randomly selected in a ratio of 4:1 from the 55,000 samples within the tactile-olfactory dataset.

The framework of the BOT associated learning architecture began with a versatile CNN for extracting tactile information from time-variant tactile mappings[17]. Because of the relatively small scale of the tactile mapping, we chose a visual geometry group (known as VGGNet) model of twice convolution to process tactile information, leading to a rapid extraction of key features about object's local topography and material stiffness[47]. This learned tactile output (512D vector) of CNN was then concatenated with the learned olfactory representation—a 512D vector of the collected olfactory data from one object—to form a new feature that served as an input to the three-layer fully connected neural network (with 0.5 dropout rate) for final learning. It is notable that the tactile and olfactory input weight during fusion process is adjusted as:

$$\begin{cases} T'_{net} = k_T \times T_{net} \\ O'_{net} = k_O \times O_{net} \\ D_T = k_D \times 512 \\ D_O = (2 - k_D) \times 512 \end{cases} \quad (1)$$

where $T'_{net}$ and $O'_{net}$ are the inputs of tactile and olfactory vectors after feature extraction; $D_T$ and $D_O$ are the length of tactile and olfactory vectors for the fusion process; $k_T$, $k_O$, and $k_D$ are the proportionality coefficients obtained from the supervised scenario-dependent feedback. Further details are in Supplementary Table 1, Supplementary Notes 5, 6, 7, and 8.

The operation of improving accuracy via changing the tactile and olfactory weight is mainly in the multimodal fusion algorithm, which is defined as:

$$I_{fusion} = MCB(T_{net}, O_{net}, n_t, n_o, d) \quad (2)$$

where $T_{net}$ and $O_{net}$ are the tactile and olfactory features extracted from CNN, $n_t$ is the length of tactile feature vector, $n_o$ is the length of olfactory feature vector and $d$ is the length of fusion feature vector. All of them are environmental parameters obtained from the supervised scenario-dependent feedback. We can rewrite above equation specifically as:

$$I_{fusion} = FFT^{-1}\left(FFT\left(\Psi\left(Resize\left(T_{net}, n_t\right)\right)\right) \odot FFT\left(\Psi\left(Resize\left(O_{net}, n_o\right)\right)\right)\right) \quad (3)$$

where FFT means fast Fourier transform, and length $(I_{fusion})$ = length $(\Psi) = d$, the Resize function adjusts the feature vector according to the proportionality coefficients of tactile and olfactory vectors in the fusion process. This operation alleviates the impact caused by interference and increases the data representativeness.

In general, the results showed that our BOT-associated learning architecture was tolerant towards defects in the input information and was better than unimodal recognition approaches.

**Development of the robotic arm system**. First, the system was developed with robot operating system (www.ros.org) using a computer. The developed system was then implemented on an industrial robot-manipulator (UR5 6-DOF) controlled by a computer through a custom-made TCP/IP communication driver. An allegro hand was then mounted on the end of the arm with full 16 DOFs and was connected to the controlling computer. Assuming a rough location was known in advance, the basic idea was to compute the trajectory of the end-effector in Cartesian space, and solve the trajectory of each DOF using inverse kinematic. In order to improve the security and accuracy of the system, the force sensors attached on the fingertips could provide force feedback information. Once the output value had exceeded a certain threshold, the system would record the attached position and replay the trajectory to avoid possible damage.

The trajectory was planned online to react instantaneously to unforeseen and unpredictable events. Online Trajectory Generation (OTG) was used in this case[48]. The basic idea of the OTG algorithm was as follows. Assuming the execution cycle of the robot is $T^{cycle}$, the time discrete overall system with a set of time instants is written as:

$$T = \{T_0, .., T_i, \dots, T_n\}, \text{with } T_i = T_i - 1 + T^{cycle} \text{ and } i \in \{1, \dots, N\} \quad (4)$$

The position of the robotic system at time, where $K$ is the number of DOFs. Velocities, accelerations, and jerks are analogously represented by $V_i$, $A_i$, and $J_i$. The current state of the motion descreted by $M_i = (p_i, v_i, A_i, J_i)$, and the motion constrains are denoted as $B_i = (V_i^{max}, A_i^{max}, J_i^{max})$. Given input parameters formated in $W_i = (M_i, M_i^{target}, B_i)$, the OTG algorithm computes the motion profile

after one cycle time:

$$Mi + 1 = f(Wi) \qquad (5)$$

We first used a decision tree to compute the synchronization time $t_{\text{sync}}$, at which all the DOFs were able to reach the target state, then another decision tree was used to determine the motion profile during this period and recomputed the trajectory for all DOFs to reach the target position simultaneously. Note that this algorithm will determine the whole motion profile from $M_i$ to $M_i^{\text{target}}$, but just execute $M_{i+1}$ at time $i$, and recomputed the trajectory at time $t + 1$ with new sensor readings.

**Statistics and reproducibility**. Each experiment was repeated at least three times independently. The experimental outcomes between independent experiments were in all cases comparable. All data are presented as mean ± standard deviation. All software used in this study for data analysis is either commercially available or open source. For example, Matlab R2020a and Origin 8.

**Reporting summary**. Further information on research design is available in the Nature Research Reporting Summary linked to this article.

## Data availability

All data needed to evaluate the conclusions in the paper are present in the paper and the Supplementary Information. Additional data related to this paper may be requested from the authors. The computational data is available from GitHub at https://github.com/wjcbob/BOT. DOI identifier: 10.5281/zenodo.5714516. year: 2021

## Code availability

The Replication code that supports the plots within this paper and other findings of this study is available from GitHub at https://github.com/wjcbob/BOT.

DOI identifier: 10.5281/zenodo.5714516. year: 2021. The code that supports the robotic arm manipulation and wireless data communication is available from the corresponding author upon reasonable request.

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

## Acknowledgements

This work was partially supported by National Science and Technology Major Project from the Minister of Science and Technology of China (grant nos. 2018AAA0103100 and 2020AAA0130100), National Science Fund for Excellent Young Scholars (grant no. 61822406), National Natural Science Foundation of China (grant nos. 61574156, 61904187 and 51703239), Shanghai Outstanding Academic Leaders Plan (grant no. 18XD1404700), Shanghai Sailing Program (grant no. 17YF1422800), Key Research Program of Frontier Sciences, CAS (grant no. ZDBS-LY-JSC024), the Strategic Priority Research Program of Chinese Academy of Science (Grant No. XDB32070203) and the Guangdong Provincial Key Research and Development Plan (Grant No. 2019B090917009).

## Author contributions

M.L., Y.Z., and J.W. contributed equally to this work. T.H.T. and M. L. conceived the idea. M. L., H.Y., and K.S. assembled the tactile data collecting system. M.L. and P.Z. assembled the olfactory data collecting system. M.L., Q.C., J.H., L.S, and J.L. built the buried scenario and performed the human detection experiment with mechanical hand. M.L. and Y.Z. performed the experiments and collected the tactile and olfactory datasets. J. W. built the algorithm. M.L., Y.Z., and J.W. analyzed the data. T.H.T., M.L., Y Z., and N.Q. prepared the manuscript. All authors discussed the results and commented on the manuscript.

## Competing interests

The authors declare no competing interests.
