## [Peer Review File · Nature Communications]

Reviewers' Comments:

Reviewer #1:

Remarks to the Author:

Teaching robots to handle general objects has historically been a challenge. As a result, there is substantial interest in new modes of sensing (along with vision) that can help robots perform this task effectively. When objects are occluded, relying solely on visual access is not suitable. Therefore, there is also high recent interest in developing tactile sensors that can help meet these demands. This current work suggests that using 14 tactile sensors at each fingertip and 6 gas sensors at the palm can help in identifying objects (specifically humans among debris).

Although the paper is written well, overall, the results follow a common strategy in the field (of combining arrays of sensors with machine learning tools to show that it is possible to identify objects). There are several fundamental points for the authors to consider in order to make the case that the current work is a new and important addition to existing literature:

1. There are many excellent examples of prior works that have already shown that objects can be identified with high accuracy using tactile sensors alone or using multiple modalities together (reviewed well in H. Liu et al. International Journal of Advanced Robotic Systems 2017). Furthermore, recent works have also shown that tactile sensing can be used to identify textures fully submerged in sand which is well beyond the level of occlusion shown in the current manuscript (e.g., R. Patel, ISER 2021 link: <https://arxiv.org/pdf/2102.10230.pdf>). The explicit combination of tactile data with gas sensing is not as common so it is critical to show why this strategy is substantially more useful (beyond what is said in Supp. Note 5) than the many combinations of modalities already published (e.g., allowing large number of objects to be distinguished using multimodal matching, J. Lin et al. ICRA 2019 or identifying human hands using tactile/temperature sensing, G. Li et al. Sci Robotics 2020 etc.).

2. The authors state in the abstract: "acquisition of the local topography, stiffness, and odor of a variety of objects without visual input". Can the authors justify this statement with specific demonstrations? (i) How is local topography obtained from a few sensors mounted at each fingertip alone: can you show the acquired topography as an example? (ii) I could not understand from the paper how stiffness is obtained here. Conceptually, to extract stiffness, both local deformation and forces are needed. Can you clarify how deformation is obtained in this case? (iii) I was curious how the sample set of 11 objects relate to the gases sensed, i.e., can you explain why it should be possible to recognize/differentiate these objects better using the specific gases detected.

Additionally, is it possible to sense gases when objects are covered with liquids like water? This could be important in wading through debris (star-nose moles have a unique capability to smell underwater). If possible, this would be useful to demonstrate.

Minor comments (partial list):

1. In all confusion matrices, please use object labels instead of a number. It would make it easier for a reader to follow.

2. There is little information on how the robot's motion is planned or the level of autonomy here. For instance, in Fig. 4h/4i, how autonomously is the robot going about removing the debris/identifying the human? I would highly encourage the authors to show videos of all these tasks since the degree of autonomy can be inferred more clearly.

3. In some studies, like tactile sensor damage in Fig. 4f/4g, did the authors include noise in the training data too by removing data from a few sensors to simulate damage? This is quite commonly done and improves the performance when few data points are lost (i.e. might help generalize the model better). In fig. 4g, while evaluating BOT-M performance, are the right amounts of gas sensors "damaged" too or is it only tactile sensor data that is removed? I believe data from the same % of gas sensors should be removed too for a fair comparison. The network design section of the supplement does not have sufficient information.

4. There are many plots in the manuscript that need error bars (or statistical measures). Are all results from one run? Please report N, std. dev. etc. as appropriate.

Reviewer #2:

Remarks to the Author:

In this work, inspired by the natural sense-fusion system of star-nose mole, Liu and co-authors reported a deep-learning assisted tactile-olfactory sensing system to achieve robust object recognition. The figures are well arranged, and the text logic is clear. However, at the technical level, the quality cannot reach the high standard of Nature Communications. The following are my comments for this manuscript:

1. Object recognition based on multimodal data have been investigated in many previously reported studies with much better performance, for example, Online Learning for Multimodal Data Fusion with Application to Object Recognition, <https://ieeexplore.ieee.org/document/8039518> and Multi-source remote sensing data fusion: status and trends, DOI: 10.1080/19479830903561035. Most importantly, there is a very similar work (Skin-inspired quadruple tactile sensors integrated on a robot hand enable object recognition, DOI: 10.1126/scirobotics.abc8134) published in 2020. In this work, the skin-inspired quadruple sensors were well designed with very high performance. Besides, there is simulations and in-depth analysis in this work (DOI: 10.1126/scirobotics.abc8134). As such, the research reported in this work is incremental by adding more modalities.

2. The authors claim that the device is capable of recognizing objects in real-time. However, the wireless protocols - latency - model calculation time is not provided.

3. All photographs in the figures are missing scale bars.

Compared with those previously reported works, the sensor device in this manuscript was incremental research. The algorithm is not new (no new insights). The variety of objects recognized is not enough, and the accuracy is competitive in the community. Therefore, in my opinion, this manuscript is not suitable for publication in Nature Communication.

Reviewer #3:

Remarks to the Author:

The authors designed a tactile-olfactory sensing array inspired by the neural system of the star nose mole. Through machine learning algorithm processing, the intelligent sensor system can accurately classify some experimental objects in some specific environments, which provides a new research idea for the design principle of intelligent sensor. However, there are still some problems with the manuscript.

1) In the introduction, the author mentioned that star-nosed moles survive in the lightless underground environment because of its tactile and olfactory perception. Should the author briefly describe some other recognition patterns in the dark environment (including biological and bionic), and put forward the advantages of tactile olfaction.

2) Figure 1d describes the olfactory tactile fusion hierarchy, and processes olfactory and tactile information with CNN. CNN is a training algorithm for recognition. How to use it for information processing?

3) There is no clear relationship between the text and the legend of Supplementary Figure 5. Its description needs to be clearer.

4) How to reduce the dimension of data to form Supplementary Figure 8? Is there any information loss?

5) Figure 3f only shows the processing results under different noises which is not in line with "while bot-m continues to maintain high recognition accuracies" in the text.

6) The Methods section should be in the past tense instead of present tense.

7) The words like "0.2 μm thick" should be replaced with "with a thickness of 0.2 μm " in the

Methods section.

8) The fabrication of 6 gas sensors should have a list of the sensing materials and the sensing principle.

9) Figure 4b should not be pointed out that ammonia has a smell similar to human body when the gas sensing principle is not put forward.

10) "the force sensor can distinguish objects with various stiffness, thus distinguishing the soft human arm from the rigid debris" is not the reason for the tactile olfactory fusion method maintains its initial high accuracy (Figure 4e)

11) It is mentioned "improve accuracy via changing the tactile and olfactory weight". Few words should be added to describe the principle and significance of this operation.

12) In Figure 4i, the rescue of surviving human beings is not secure, because in reality, people are not only partially buried. It is recommended to delete this figure.

13) The resulting outcome is very significant for the development of robust object recognition in harsh environments. Unfortunately, the writing ability of this manuscript is approximately terrible. As a scientific article, the main text should focus on the in-depth mechanism instead of experimental phenomenon description. For example, the force sensing mechanism, as well as the realization method of the susceptible gas sensor with specificity in this work, should be detailedly discussed. Worst of all, the result of the BOT-M with highest recognition rate is also with no interpretation. Here, I strongly recommend adding these discussion.

14) What is the unique gas of human beings? Could this design discriminate living person from dead bodies? Animal should be added as interference objects because it is the most human-like one.

15) The following related references, which have been published recently, could be cited to highlight the significance of developing multimode sensing array with bionic strategy (Advanced Materials, 2020, 32, 1907043).

16) In order to consolidate the excellent property of robotic arm and make the main purpose of this article unambiguous, the video related with Figure 4h should be added as supporting file. In this manuscript, a new construction mode of intelligent sensor with tactile-olfactory recognition is designed, and its performance is systematically characterized to verify the principle of synergistic enhancement effect. There is a lack of explanation of methods used in the study and some descriptions are not rigorous enough. In addition, there are problems with sentence structure, verb tense, and clause construction in this manuscript. I suggest that this manuscript should be reconsidered after major revision.

Point by point response (comments in black and responses are in blue):

Reviewer #1:

Teaching robots to handle general objects has historically been a challenge. As a result, there is substantial interest in new modes of sensing (along with vision) that can help robots perform this task effectively. When objects are occluded, relying solely on visual access is not suitable. Therefore, there is also high recent interest in developing tactile sensors that can help meet these demands. This current work suggests that using 14 tactile sensors at each fingertip and 6 gas sensors at the palm can help in identifying objects (specifically humans among debris).

Although the paper is written well, overall, the results follow a common strategy in the field (of combining arrays of sensors with machine learning tools to show that it is possible to identify objects). There are several fundamental points for the authors to consider in order to make the case that the current work is a new and important addition to existing literature:

Response:

We would like to express our sincere thanks to the reviewer for positively evaluating that our reported results of objects recognition (specifically humans among debris) using the tactile-olfactory sensing system. Each of the following comments is highly important to improve the quality of our manuscript. The followings are the point-to-point replies to the comments.

1. There are many excellent examples of prior works that have already shown that objects can be identified with high accuracy using tactile sensors alone or using multiple modalities together (reviewed well in H. Liu et al. International Journal of Advanced Robotic Systems 2017). Furthermore, recent works have also shown that tactile sensing can be used to identify textures fully submerged in sand which is well beyond the level of occlusion shown in the current manuscript (e.g., R. Patel, ISER 2021 link: <https://arxiv.org/pdf/2102.10230.pdf>). The explicit combination of tactile data with gas sensing is not as common so it is critical to show why this strategy is substantially more useful (beyond what is said in Supp. Note 5) than the many combinations of modalities already published (e.g., allowing large number of objects to be distinguished using multimodal matching, J. Lin et al. ICRA 2019 or identifying human hands using tactile/temperature sensing, G. Li et al. Sci Robotics 2020 etc.).

Response:

We sincerely thank the reviewer for the insightful suggestion. We agree that there have been certain researches on similar aspects in recent years and each of them has some advantages. However, **the system in our article contains its own superiorities**. Here, we summary a comparison table about the recent works in objects recognition by solely tactile sensors and by multiple modalities together; we compared various sensing methods by highlighting several key indicators, including data size, methods, accuracy and specialties (**Table R1**).

Odor is the chemical fingerprint of every object (DOI: 10.3389/fpsyg.2014.00504), but

until now has rarely been combined with tactile sensing for object identification. As the reviewer said, many recognition modalities already published do not use the explicit combination of tactile and olfactory sensing, mainly because gas sensors have both dimensional and temporal data mismatch with force sensors, and are susceptible to ambient gas interference. Nevertheless, by the effective data preprocessing and complementary setting of 6 different gas sensors, **we have proved that olfactory sensing can also be suitable for feature fusion with tactile data.** Therefore, the olfactory sensing provides an alternative option for object recognition, which essentially carries more information of objects compared to other ordinary physical parameters, like the temperature as the reviewer mentioned.

Comparing with the vision-based sensing systems, our tactile and olfactory fusion strategy has relatively small input data size, leading to smaller requirements of computing resource and faster identifications, which are crucial in rescue mission. In addition, for recent studies about objects identification using solely tactile sensors, as mentioned by the reviewer, most of them choose to use **flexible none-silicon based strain sensors.** However, in our design, **we use silicon-based force sensors fabricated by micro-electromechanical systems (MEMS) technologies, which have more robust performance, smaller size, and higher accuracy.** Meanwhile, comparing to the existed tactile-based sensing systems, **our strategy for the first time takes odor as one input modality and thus is more suitable for situations that gas plays an important role,** such as human recognition in rescue scenarios.

Furthermore, as mentioned by the reviewer, there are also some other works using tactile sensing to identify surface texture of object fully submerged in sand. However, in their work, the exact location of the tested object in the sand should be known in advance in order to achieve the following sensing performance. Also, only depending on texture identification is not enough for objects differentiation when these objects have similar covering materials or similar texture. Thus, it's necessary to identify objects from different modalities to achieve higher accuracy. In our work, **the tactile sensing can provide information of local topography (roughness) and material stiffness; the olfactory sensing can offer odor information of the tested objects.** After a combination of these aspects, a more reliable object recognition can be achieved, providing concrete support for potential rescue mission.

We have added corresponding discussion in the revised manuscript (Please see Page 12, Table 2, Discussion and Supporting Information).

Table R1. Comparison of different object recognition methods.

	Sensor type	Algorithm	Input size	Data size	Computing resource	Method	Multi-modality	Accuracy	Specialty	Deficiency	Ref.
Need Imaging	Depth camera	Dex-Net4.0	—	Large	Large	Depth image	No	95%	—	Need direct and clear visualization	[1]

	GelSight & RGB camera	ResNet50	128×128×3+256×256×3	Medium	Medium	Grasp & image	Yes	—	—	Need direct and clear visualization & unsuitable for varied grasping	[3]
	Stretchable strain & camera	Alex-Net	160×120+5	Light	Medium	Somatosensory & image	Yes	100%	Improved accuracy under dim light	Need large data size and computing resource	[7]
	GelSight	ResNet50	—	Medium	Medium	Dig(touch)	No	99%	Recognizing object submerged in sand	Need specific object & deficient anti-interference	[2]
	Tactile	ResNet18	32×32	Medium	Medium	Grasp	No	>90%	Learning the grasping pattern	Need manual grasping	[8]
	Haptic stimulator	CNN, SVM	16×200	Medium	Light	Grasp	Yes	96%	Providing haptic-feedback for human-machine interface	Need specific object & deficient anti-interference	[4]
Without Imaging	Quadruple tactile	MLP	4×10	Light	Light	Grasp	Yes	94%	Suitable for robotic hand	Limited workable environment & deficient anti-interference	[5]
	Pressure&vibration	ANN	—	Medium	Light	Touch	Yes	99.1%	Recognizing surface texture	Need specific object & limited algorithm	[6]
	Tactile & olfactory	BOT (CNN/FCN/ MCB)	5×14+6	Light	Light	Touch	Yes	96.9%	Improved anti-interference & Suitable for rescue scenarios	Fully underwater environment	This work

2. The authors state in the abstract: “acquisition of the local topography, stiffness, and odor of a variety of objects without visual input”. Can the authors justify this statement with specific demonstrations?

Response:

Thanks to the reviewer for this suggestion. We totally agree that these detailed demonstrations mentioned by the reviewer contain high value. **Here, we have added supplementary experiments to demonstrate these missing features, including local topography, stiffness characterization, and odors of various objects.**

(i) How is local topography obtained from a few sensors mounted at each fingertip alone: can you show the acquired topography as an example?

Response:

Thanks to the reviewer for this question. Here, we make supplementary characterizations of tactile perception of object’s surface texture based on the tactile sensing array (1*14) mounted at one fingertip alone. As shown in **Figure R1** (a), height gradient with each step around 0.1 mm could be built by stacking various amount of A4 paper together; the single tactile sensing array (1*14) is used to perceive the existence of the paper. As shown in Figure R1 (b), the minimum

detectable height difference for our tactile sensing array is around 0.3 mm, while larger height difference (0.3 to 1.5 mm) can be differentiated. Thus, the results of this experiment prove our tactile sensing array can perceive local topography with limit of detection of 0.3 mm.

Figure R1. Tactile perception of height gradient. a) Schematic illustration of using single tactile sensing array (1*14) to detect height gradient. b) Output voltage of force sensors while touching paper pile with thickness from 0.2mm to 1.5mm.

Furthermore, to prove the local topography of the more practical object can be perceived from a few sensors mounted at each fingertip alone, we used the tactile sensing arrays (5*14) mounted on the fingertips of a mechanical hand to detect the roughness of apple and orange surfaces. As shown in **Figure R2** (b) and (d), single tactile sensing array can accurately present the surface roughness differentiation between the tested objects. Thus, the local topography of tested objects can be obtained from the five tactile sensing arrays mounted at each fingertips of a mechanical hand (Figure R2 (a) and (c)).

We have added corresponding discussion in the revised manuscript (Please see Page 6, 34 and 35, Note 2, Supplementary Figure 9, 10, Supporting Information).

Figure R2. Tactile perception of different objects with various surface roughness. a) The tactile mapping of the local topography of an apple surface, detected by the tactile sensing arrays (5*14) mounted on the fingertips of a mechanical hand. b) Output voltage of every force sensor from a single tactile sensing array (1*14), which present the local roughness of the apple surface. c) The tactile mapping of the local topography of an orange surface, detected by the same tactile sensing arrays (5*14). d) Output voltage of every force sensor from a single tactile sensing array (1*14), which present the local roughness of the orange surface.

(ii) I could not understand from the paper how stiffness is obtained here. Conceptually, to extract stiffness, both local deformation and forces are needed. Can you clarify how deformation is obtained in this case?

Response:

Thanks to the reviewer for pointing out this point. We totally agree that both force and local deformation are needed to extract stiffness. Here, **Figure R3** (a) shows the sensitive hexagonal silicon membrane within the force sensor, on which located a Wheatstone bridge made by four piezoresistors. In principle, when the force sensor is in contact with the detected object, the membrane would form a certain deformation, the degree of which depends on both the stiffness of the object and the force applied. Then the membrane deformation would cause the changes in resistance values of the four piezoresistors. Thus, the voltage output value of the Wheatstone bridge could reflect the change of resistance value and the degree of membrane deformation.

Based on this working principle, **when we use the mechanical hand to apply the same amount of force to touch the object with uniform topography, the output voltage value would be positively correlated to the local stiffness.** As shown in Figure R2 (b) and (d), because the stiffness of apple is relatively larger than orange, the average output voltage value of the sensor array for apple (around 5V) is larger than which for orange (around 2.5V). This result

proves the capability of our tactile sensing arrays to distinguish the stiffness of various objects, providing important information for the following object recognition process.

We have added corresponding discussion in the revised manuscript (Please see Page 6, 35 and 36, Note 2 and 3, Supplementary Figure 3, Supporting Information).

Figure R3. Interior structure of a single force sensor. a) SEM image showing the sensitive hexagonal silicon membrane within the force sensor, on which located a Wheatstone bridge made by four piezoresistors. b) Schematic illustration of the electronic circuit of the four piezoresistors.

(iii) I was curious how the sample set of 11 objects relate to the gases sensed, i.e., can you explain why it should be possible to recognize/differentiate these objects better using the specific gases detected.

Response:

Thanks to the reviewer for this question. In order to provide a clear explanation and avoid any confusion to the readers, we have added the information of working principle of our gas sensors in the revised manuscript. (Please see Page 15, 36 and 37)

The working principle of the gas sensor is similar to that of the conventional semiconductor resistive sensor. While placing the sensors in the certain gas environment, the detected gas molecules would combine to the surface of the sensitive semiconductor material and a corresponding chemical reaction would occur. In this process, the electron transfer produced by the chemical reaction could cause the resistance changes of the semiconductor material. By measuring the resistance change of the material, the status of the gas detection could be known. However, these reactions are not absolutely specific, which means instead of responding to a certain type of gas molecule, one sensor could respond to multiple gases in different responding degree. For example, the sensor number one would react to various gases, but only has the highest response to ethanol. Therefore, through multiple experiments, we can know the corresponding reaction sensitivity of these materials to different gas molecules. The gas types, including ethanol, acetone, ammonia, carbon monoxide, hydrogen sulfide and methane, as we

mentioned in the article, are the most responsive gases corresponding to the six materials according to the experiment results. More importantly, **since the sensitivities of each material to different gas molecules are different, we select the complementary setup (6 gas sensors) to sense the type and concentration of various gases, reflecting the odor profile of the detected object.**

In this article, our intention is to differentiate the odors of human being and other common objects in order to achieve human recognition in the burial scenarios. Therefore, we choose the mentioned 11 objects, including orange, towel, stone, can, worn clothes, carton, mug, mouse, hair, leg and arm in the experiments of this article. The results (Figure 2h) show that our six-channel gas sensing array could accurately capture the unique odor profiles of those objects.

Additionally, is it possible to sense gases when objects are covered with liquids like water? This could be important in wading through debris (star-nose moles have a unique capability to smell underwater). If possible, this would be useful to demonstrate.

Response:

Thanks to the reviewer for this question. We totally agree that olfactory sensing for objects covered with liquids like water could be important in wading through debris, which is common in real rescue scene. **Our olfactory sensor array can sense gases when objects are covered with water.** Here we make supplementary characterizations of these missing features for tactile and olfactory sensing.

As shown in **Figure R4**, we use oranges (normal, covered with water and muddy water) as the target objects for identification. During this process, because of the encapsulation, the olfactory sensing array has a certain capability to withstand water environment, which is sufficient to detect objects covered with water or muddy water without causing any damage to sensors. However, for long-term underwater immersion (≥ 1 day), our current sensor packaging method might not be sufficient and more improvements on this problem can be carried on in our future research.

Compared to the dry condition, when the orange is covered with water, the response degree of each olfactory sensor has been reduced slightly, because the increased humidity affects the responsive reaction. For the situation when orange is covered with muddy water, with the interference of the mud odor, the responses are further affected. However, **while combining olfactory sensing with tactile perception, the interference of olfactory perception can be alleviated and a relative high recognition accuracy can be maintained whether the target object is cover with water or muddy water, which also demonstrates the outstanding anti-interference ability of the tactile-olfactory sensing system in our work.**

Figure R4. Olfactory and tactile perception of oranges being covered with water and muddy water. a-c) Images of olfactory and tactile perception of orange in normal, covered with water, and covered with muddy water. d) The responses of six gas sensors when sensing orange covered with water and muddy water. e) The recognition accuracy of orange covered with water and muddy water using olfactory-based recognition and BOT-M associated learning.

We have added corresponding discussion in the revised manuscript (Please see Page 6, 37 and 38, Note 5, Supplementary Figure 11, Supporting Information).

Minor comments:

1. In all confusion matrices, please use object labels instead of a number. It would make it easier for a reader to follow.

Response:

We sincerely thank the reviewer for this insightful suggestion. We have made some improvement to help the reader to follow. Since full-name object labels could be illegible in the picture, we added the abbreviations of the objects in the picture and full-name explanations in the caption as shown in the Figure R5 (a) and (b).

We have added corresponding correction in the revised manuscript (Please see Page 30, 50, Figure 3, and Supplementary Figure 15, Supporting Information).

Figure R5. Confusion matrix of the sole tactile recognition strategy (a) and the BOT-M recognition strategy (b). The full name of the abbreviation: Org-Orange, Twl-Towel, Arm-Arm, Stn-Stone, Can-Can, Hir-Hair, Leg-Leg, Ms-Mouse, Clth-Worn Clothes, Mug-Mug, Ctn-Carton.

2. There is little information on how the robot’s motion is planned or the level of autonomy here. For instance, in Fig. 4h/4i, how autonomously is the robot going about removing the debris/identifying the human? I would highly encourage the authors to show videos of all these tasks since the degree of autonomy can be inferred more clearly.

Response:

We sincerely thank the reviewer for this insightful suggestion. In order to show the autonomy level of the robotic motion more clearly, we have recorded a video about various tasks mentioned in Fig 4h/4i, including **Removing debris and Human detection by the mechanical hand**. As shown in the video, we mounted the mechanical hand on a robotic arm to recognize and grab the debris, and then move them away from the designated area; then we let the same robotic system to touch and recognize the fully-exposed human arm/leg.

We have added corresponding discussion in the revised manuscript (Please see Page 11 and Supporting Information).

3. In some studies, like tactile sensor damage in Fig. 4f/4g, did the authors include noise in the training data too by removing data from a few sensors to simulate damage? This is quite commonly done and improves the performance when few data points are lost (i.e. might help generalize the model better).

Response:

We sincerely thank the reviewer for this insightful suggestion. There could be a good choice in practical application scenarios. But in this article, we did not apply the same treatment to the training data by removing data from a few sensors to simulate the environmental interference. Although this operation might help generalize the model better, **we afraid that this approach would cause the test set and the training set to have an artificially high similarity, causing inaccuracy in recognition results and failing to reflect the actual performance of the model**. Thus, in the experiment, we modify tactile training data sets by using other common processing

methods, including translation and flipping, to generalize the model better and get a satisfactory result of recognition.

In fig. 4g, while evaluating BOT-M performance, are the right amounts of gas sensors “damaged” too or is it only tactile sensor data that is removed? I believe data from the same % of gas sensors should be removed too for a fair comparison. The network design section of the supplement does not have sufficient information.

Response:

Thanks to the reviewer for this question. We totally agree that data from the same % of gas sensors should be removed too for a fair comparison in Fig. 4g. Here, we make supplementary experiment of removing both of tactile sensors and gas sensors, the performance of BOT-M is still outstanding as shown in **Figure R6** (a) and (b).

We have added corresponding change in the revised manuscript (Please see Page 11 and Figure 4 (f), (g)).

Figure R6. Human recognition when tactile and olfactory sensors has been partially damaged.

a) Schematic showing random part of the sensors in the tactile and olfactory arrays being damaged. b) The accuracy of arm recognition with different sensor damage rate using tactile-based recognition and BOT-M associated learning.

4. There are many plots in the manuscript that need error bars (or statistical measures). Are all results from one run? Please report N, std. dev. etc. as appropriate.

Response:

We sincerely thank the reviewer for this insightful suggestion. We have added those missing features in the manuscript. Each result in these plots was from ten runs. The height of bars presents the average value of these ten runs results and the error bars shows the standard deviation of these results.

We have added corresponding discussion in the revised manuscript (Please see Page 31-34, Figure 3 (d) and Figure 4 (g)).

Figure R7. Error bars in the plots within the manuscript. a) BOT-M associated learning shows the best accuracy among the unimodal (tactile and olfactory) and multimodal fusion strategies (BOT, BOT-R, BOT-F, BOT-M). b) The accuracy of arm recognition with different sensor damage rate using tactile-based recognition and BOT-M associated learning. $n = 10$.

Reviewer #2:

In this work, inspired by the natural sense-fusion system of star-nose mole, Liu and co-authors reported a deep-learning assisted tactile-olfactory sensing system to achieve robust object recognition. The figures are well arranged, and the text logic is clear. However, at the technical level, the quality cannot reach the high standard of Nature Communications. The following are my comments for this manuscript:

Response:

We would like to express our genuine thanks to the reviewer for evaluating that the figures are well arranged, and the text logic is clear. The careful comments from this reviewer helped us further improve the quality and clarity of the manuscript. The followings are our replies to each of the comments.

1. Object recognition based on multimodal data have been investigated in many previously reported studies with much better performance, for example, Online Learning for Multimodal Data Fusion with Application to Object Recognition, <https://ieeexplore.ieee.org/document/8039518> and Multi-source remote sensing data fusion: status and trends, DOI: 10.1080/19479830903561035. Most importantly, there is a very similar work (Skin-inspired quadruple tactile sensors integrated on a robot hand enable object recognition, DOI: 10.1126/scirobotics.abc8134) published in 2020. In this work, the skin-inspired quadruple sensors were well designed with very high performance. Besides, there is simulations and in-depth analysis in this work (DOI: 10.1126/scirobotics.abc8134). As such, the research reported in this work is incremental by adding more modalities.

Response:

We sincerely thank the reviewer for providing us with these high-quality references. We have thoroughly read them and found them influential. We agree that there have been certain researches on similar aspects in recent years and each of them has some advantages. However, **the system in our article contains its own superiorities**. Here, we summary a comparison table about the recent works in objects recognition by solely tactile sensors and by multiple modalities together; we compared various sensing methods by highlighting several key indicators, including data size, methods, accuracy and specialties (**Table R2**).

Odor is the chemical fingerprint of every object (DOI: 10.3389/fpsyg.2014.00504), **but until now has rarely been combined with tactile sensing for object identification**. Many recognition modalities already published do not use the explicit combination of tactile and olfactory sensing, mainly because gas sensors have both dimensional and temporal data mismatch with force sensors, and are susceptible to ambient gas interference. Nevertheless, by the effective data preprocessing and complementary setting of 6 different gas sensors, **we have proved that olfactory sensing can also be suitable for feature fusion with tactile data**. Therefore, the olfactory sensing provides an alternative option for object recognition, which essentially carries more information of objects compared to other ordinary physical parameters,

like the temperature as the reviewer mentioned (quadruple sensors).

Comparing with the vision-based sensing systems, our tactile and olfactory fusion strategy has relatively small input data size, leading to smaller requirements of computing resource and faster identifications, which are crucial in rescue mission. In addition, for recent studies about objects identification using solely tactile sensors, most of them choose to use **flexible none-silicon based strain sensors**. However, in our design, **we use silicon-based force sensors fabricated by micro-electromechanical systems (MEMS) technologies, which have more robust performance, smaller size, and higher accuracy**. Meanwhile, comparing to the existed tactile-based sensing systems, **our strategy for the first time takes odor as one input modality and thus is more suitable for situations that gas plays an important role**, such as human recognition in rescue scenarios.

Furthermore, there are also some other works using tactile sensing to identify surface texture of object fully submerged in sand. However, in their work, the exact location of the tested object in the sand should be known in advance in order to achieve the following sensing performance. Also, only depending on texture identification is not sufficient for objects differentiation when these objects have similar covering materials or similar texture. Thus, it's necessary to identify objects from different modalities in order to achieve higher accuracy. In our work, **the tactile sensing can provide information of local topography (roughness) and material stiffness; the olfactory sensing can offer odor information of the tested objects**. After a combination of these aspects, a more reliable object recognition can be achieved, providing concrete support for potential rescue mission.

We have added corresponding discussion in the revised manuscript (Please see Page 12, Table 2, Discussion and Supporting Information).

Table R2. Comparison of different object recognition methods.

	Sensor type	Algorithm	Input size	Data size	Computing resource	Method	Multi-modality	Accuracy	Specialty	Deficiency	Ref.
Need Imaging	Depth camera	Dex-Net4.0	—	Large	Large	Depth image	No	95%	—	Need direct and clear visualization	[1]
	GelSight & RGB camera	ResNet50	128×128×3+256×256×3	Medium	Medium	Grasp & image	Yes	—	—	Need direct and clear visualization & unsuitable for varied grasping	[3]
	Stretchable strain & camera	Alex-Net	160×120+5	Light	Medium	Somatosensory & image	Yes	100%	Improved accuracy under dim light	Need large data size and computing resource	[7]
Without Imaging	GelSight	ResNet50	—	Medium	Medium	Dig(touch)	No	99%	Recognizing object submerged in sand	Need specific object & deficient anti-interference	[2]
	Tactile	ResNet18	32×32	Medium	Medium	Grasp	No	>90%	Learning the grasping pattern	Need manual grasping	[8]

Haptic stimulator	CNN, SVM	16×200	Medium	Light	Grasp	Yes	96%	Providing haptic-feedback for human-machine interface	Need specific object & deficient anti-interference	[4]
Quadruple tactile	MLP	4×10	Light	Light	Grasp	Yes	94%	Suitable for robotic hand	Limited workable environment & deficient anti-interference	[5]
Pressure&vibration	ANN	—	Medium	Light	Touch	Yes	99.1%	Recognizing surface texture	Need specific object & limited algorithm	[6]
Tactile & olfactory	BOT (CNN/FCN/ MCB)	5×14+6	Light	Light	Touch	Yes	96.9%	Improved anti-interference & Suitable for rescue scenarios	Fully underwater environment	This work

2. The authors claim that the device is capable of recognizing objects in real-time. However, the wireless protocols - latency - model calculation time is not provided.

Response:

We sincerely thank the reviewer for this insightful suggestion. In order to show the real-time object recognition processing, including model calculation time and the latency, we have added the screen-recording video of our algorithm. As shown in the **Figure R8** and SI video 1 and 2, we can recognize the objects within seconds. In addition, we have also recorded a video (SI video 1 and 2) about the Removing debris and Human detection by the mechanical hand, which demonstrate the operation procedures of the system for the human recognition under debris.

Figure R8. Screenshot of the Software User Interface video, showing the recognition process.

3. All photographs in the figures are missing scale bars.

Response:

We sincerely thank the reviewer for this insightful suggestion. We have added the scale bars to all the figures in the revised manuscript (Please see Page 27-34, Figure 1-4 and Supporting Information).

Reviewer #3:

The authors designed a tactile-olfactory sensing array inspired by the neural system of the star nose mole. Through machine learning algorithm processing, the intelligent sensor system can accurately classify some experimental objects in some specific environments, which provides a new research idea for the design principle of intelligent sensor. However, there are still some problems with the manuscript.

Response:

We would like to express our sincere thanks to the reviewer for positively evaluating that our tactile-olfactory sensing system provides innovative solution for object recognition, specifically for human rescue mission using based mechanical hand. Each of the following comments is highly important to improve the quality of our manuscript. The followings are the point-to-point replies to the comments.

1) In the introduction, the author mentioned that star-nosed moles survive in the lightless underground environment because of its tactile and olfactory perception. Should the author briefly describe some other recognition patterns in the dark environment (including biological and bionic), and put forward the advantages of tactile olfaction.

Response:

Thanks to the reviewer for this question. In nature, many animals mainly live in relatively dark environments and thus have developed multiple special sensing abilities in order to survive under obstructed vision. **Table R3** shows a comparison table about the recognition patterns of animals in nature that live in the dark environment.

In the table, we compared various sensing methods of different animals, including bats, octopus, snakes, owls, platypus, catfish, cavefish and mantis shrimp, highlighting features such as sensing methods and suitable scenarios. From the table, it's clear to find out that for some of these animals, such as bats, snakes, platypus, catfish and cavefish, they strengthen a certain sensing modality in order to increase their viability when their vision is obstructed. However, for other animals, including octopus, owl, mantis shrimp, and **star-nosed mole** from this work, they developed multimodal sensing abilities. Notably, most of the animals choose one of the senses, such as hearing and tactile sensing, to assist vision perception in the dark environment, while **star-nosed mole has relatively none vision and solely depends on tactile and olfactory perception.**

For most animals, the main function of their vision-based multimodal perception is to locate the preying target during the predation process. However, for the unique star-nosed mole, they prey by sensing the objects' texture and odor to distinguish whether the target is edible. This specific evolution is largely due to the narrow and almost dark living environment (underground) of the star-nosed mole. Thus, the main advantage of tactile-olfactory fusion is to ensure a more detailed and accurate perceptual recognition of objects in those specific environments. This

situation is similar to the rescue scenario, where human could be buried underground. Therefore, inspired by the star-nosed mole, we choose the tactile-olfactory bionic sensing array for robust object recognition in non-visual environments.

Table R3. Comparison of different recognition patterns of animals in the dark environment.

Animal name	Class	Special Sensing	Biological organs	Suitable scenario	Reference
Bats	Mammal	Ultrasound	Throat/Ears	Location orientation	[1]
Octopus	Reptile	Light/Tactile	Skin	Camouflage/Prey	[2]
Snakes	Cephalopod	Infrared Radiation	Infrared receptor	Prey	[3,4]
Owl	Aves	Light/Sound	Eyes/Ears	Prey	[5]
Platypus	Mammal	Electroreception	Mucous glands	Prey/Underwater orientation	[6]
Catfish	Pisces	Taste	Whiskers	Prey	[7]
Cavefish	Pisces	Sound	Lateral line system	Prey	[8]
Mantis Shrimp	Malacostraca	Visible/ultraviolet light	Visual System	Prey	[9]
Star-nosed mole	Mammal	Tactile/Olfactory	Tentacles/nostrils	Prey/Objects recognition	[10-12]

For bionic/artificial object recognition, we summarize a comparison table about the recent works in objects recognition by using tactile sensors alone or using multiple modalities together; we compared various sensing methods by highlighting several key indicators, including data size, methods, accuracy and specialties (**Table R4**).

Odor is the chemical fingerprint of every object (DOI: 10.3389/fpsyg.2014.00504), **but until now has rarely been combined with tactile sensing for object identification**. Many recognition modalities already published do not use the explicit combination of tactile and olfactory sensing, mainly because gas sensors have both dimensional and temporal data mismatch with force sensors, and are susceptible to ambient gas interference. Nevertheless, by the effective data preprocessing and complementary setting of 6 different gas sensors, **we have proved that olfactory sensing can also be suitable for feature fusion with tactile data**. Therefore, the olfactory sensing provides an alternative option for object recognition, which essentially carries more information of objects compared to other ordinary physical parameters, like the temperature as the reviewer mentioned (quadruple sensors).

Comparing with the vision-based sensing systems, our tactile and olfactory fusion strategy has relative small input data size, leading to smaller requirements of computing resource and faster identifications, which are crucial in rescue mission. In addition, for recent studies about objects identification using solely tactile sensors, most of them choose to use **flexible none-silicon based strain sensors**. However, in our design, **we use silicon-based force sensors fabricated by micro-electromechanical systems (MEMS) technologies, which have more robust performance, smaller size, and higher accuracy**. Meanwhile, comparing to the existed tactile-based sensing systems, **our strategy for the first time takes odor as one input modality and thus is more suitable for situations that gas plays an important role**, such as human

recognition in rescue scenarios.

Furthermore, there are also some other works using tactile sensing to identify surface texture of object fully submerged in sand. However, in their work, the exact location of the tested object in the sand should be known in advance in order to achieve the following sensing performance. Also, only depending on texture identification is not sufficient for objects differentiation when these objects have similar covering materials or similar texture. Thus, it's necessary to identify objects from different modalities in order to achieve higher accuracy. In our work, **the tactile sensing can provide information of local topography (roughness) and material stiffness; the olfactory sensing can offer odor information of the tested objects.** After a combination of these aspects, a more reliable object recognition can be achieved, providing concrete support for potential rescue mission.

We have added corresponding discussion in the revised manuscript (Please see Page 12, Table 2, 3, Discussion and Supporting Information)..

Table R4. Comparison of different object recognition methods.

	Sensor type	Algorithm	Input size	Data size	Computing resource	Method	Multi-modality	Accuracy	Specialty	Deficiency	Ref.
Need Imaging	Depth camera	Dex-Net4.0	—	Large	Large	Depth image	No	95%	—	Need direct and clear visualization	[1]
	GelSight & RGB camera	ResNet50	128×128×3+256×256×3	Medium	Medium	Grasp & image	Yes	—	—	Need direct and clear visualization & unsuitable for varied grasping	[3]
	Stretchable strain & camera	Alex-Net	160×120+5	Light	Medium	Somatosensory & image	Yes	100%	Improved accuracy under dim light	Need large data size and computing resource	[7]
Without Imaging	GelSight	ResNet50	—	Medium	Medium	Dig(touch)	No	99%	Recognizing object submerged in sand	Need specific object & deficient anti-interference	[2]
	Tactile	ResNet18	32×32	Medium	Medium	Grasp	No	>90%	Learning the grasping pattern	Need manual grasping	[8]
	Haptic stimulator	CNN, SVM	16×200	Medium	Light	Grasp	Yes	96%	Providing haptic-feedback for human-machine interface	Need specific object & deficient anti-interference	[4]
	Quadruple tactile	MLP	4×10	Light	Light	Grasp	Yes	94%	Suitable for robotic hand	Limited workable environment & deficient anti-interference	[5]
	Pressure&vibration	ANN	—	Medium	Light	Touch	Yes	99.1%	Recognizing surface texture	Need specific object & limited algorithm	[6]
	Tactile & olfactory	BOT (CNN/FCN/ MCB)	5×14+6	Light	Light	Touch	Yes	96.9%	Improved anti-interference & Suitable for rescue scenarios	Fully underwater environment	This work

2) Figure 1d describes the olfactory tactile fusion hierarchy, and processes olfactory and tactile information with CNN. CNN is a training algorithm for recognition. How to use it for information processing?

Response:

Thanks to the reviewer for pointing out this point. The convolutional neural network (CNN) is a class of artificial neural network, most commonly applied to analyze visual imagery. This is because CNN also can conduct automatic feature extraction, to generate an abstract generic data representation from the primary dataset with diverse dimensions (doi:10.1038/nature14539).

Following the same principle, the tactile and olfactory information collected by our sensors is sorted into **parallel tactile image and olfactory sequence** after data preprocessing. We analyze the feature vectors extracted by CNN, fuse them by MCB, and then use a fully connected neural network to complete the classification. Therefore, the CNN provides key features for the following algorithm, as an important and effective method for information processing.

We have added corresponding discussion in the revised manuscript (Please see Page 39, Note7, Supporting Information).

3) There is no clear relationship between the text and the legend of Supplementary Figure 5. Its description needs to be clearer.

Response:

Thanks to the reviewer for pointing out this confusing point. Previous Supplementary Figure S5 (current Supplementary Figure 6 in the revised manuscript) mainly presents the stability of our olfactory sensors. Note that the performance of gas sensors could be affected by environmental changes, causing data drifting after long-term usage. Thus, it is important for them to maintain steady during the actual detecting process. As shown in Figure R9, the performance of our gas sensor keeps steady over 60 days, providing accurate olfactory dataset for object recognition.

We have added corresponding discussion in the revised manuscript We have added corresponding discussion in the revised manuscript (Please see Page 6).

Figure R9. Stable performance of gas sensor in air (red line) and in 100 ppm ethanol

(blue line).

4) How to reduce the dimension of data to form Supplementary Figure 8? Is there any information loss?

Response:

Thanks to the reviewer for pointing out this point. The t-distributed stochastic neighbor embedding (t-SNE) is a technique that visualizes high-dimensional data by giving each data point a location in a two or three-dimensional map, which was adopted by many researchers and articles. Nat. Electron. 3, 563-570, Nature 569, 698-702, J. Mach.Learn. Res. 9, 2579–2605.

We use the same t-SNE only for visualizing tactile and olfactory data, and not for information analyzation. Therefore, although there is a certain degree of information loss, it will not affect the object recognition results.

5) Figure 3f only shows the processing results under different noises which is not in line with "while bot-m continues to maintain high recognition accuracies" in the text.

Response:

Thanks to the reviewer for this comment. In Figure 3f, compared with pure tactile and pure olfactory strategies (with accuracy around 50% and 30%), BOT-M has higher anti-noise ability (with accuracy around 90%) under different degrees of noise attack. When both tactile and olfactory data have noisy, although the accuracy is affected (around 70%), it still has clear advantages in comparison with other algorithms. Therefore, we conclude that BOT-M continues to maintain high recognition accuracies (Figure 3f).

6) The Methods section should be in the past tense instead of present tense.

Response:

Thanks to the reviewer for pointing out this problem. We have corrected this mistake in the revised manuscript. (Please see Page 13-19)

7) The words like "0.2 μm thick" should be replaced with "with a thickness of 0.2 μm " in the Methods section.

Response:

Thanks to the reviewer for pointing out this problem. We have corrected this mistake in the revised manuscript. (Please see Page 13-14)

8) The fabrication of 6 gas sensors should have a list of the sensing materials and the sensing principle.

Response:

Thanks to the reviewer for this suggestion. We have added information about the sensing materials and the sensing principle in the fabrication of gas sensors. (Please see Page 15, 37 and 38)

The sensing materials of these 6 complementary gas sensors are: (1) carbon nanotubes modified by magnesium oxide particles; (2) carbon nanotubes modified with platinum particles;

(3) graphene modified with copper oxide particles; (4) platinum-doped tin oxide; (5) platinum-doped tungsten oxide; (6) composite material of zinc oxide and tin oxide.

The working principle of the gas sensor is similar to that of the conventional semiconductor resistive sensor. While placing the sensors in the certain gas environment, the detected gas molecules would combine to the surface of the sensitive semiconductor material and a corresponding chemical reaction would occur. In this process, the electron transfer produced by the chemical reaction could cause the resistance changes of the semiconductor material. By measuring the resistance change of the material, the status of the gas detection could be known. However, these reactions are not absolutely specific, which means instead of responding to a certain type of gas molecule, one sensor could respond to multiple gases in different responding degree. For example, the sensor number one would react to various gases, but only has the highest response to ethanol. Therefore, through multiple experiments, we can know the corresponding reaction sensitivity of these materials to different gas molecules. The gas types, including ethanol, acetone, ammonia, carbon monoxide, hydrogen sulfide and methane, as we mentioned in the article, are the most responsive gases corresponding to the six materials according to the experiment results. More importantly, **since the sensitivities of each material to different gas molecules are different, we select the complementary setup (6 gas sensors) to sense the type and concentration of various gases, reflecting the odor profile of the detected object.**

In this article, our intention is to differentiate the odors of human being and other common objects in order to achieve human recognition in the burial scenarios. Therefore, we choose the mentioned 11 objects, including orange, towel, stone, can, worn clothes, carton, mug, mouse, hair, leg and arm in the experiments of this article. The results (Figure 2h) show that our six-channel gas sensing array could accurately capture the unique odor profiles of those objects.

9) Figure 4b should not be pointed out that ammonia has a smell similar to human body when the gas sensing principle is not put forward.

Response:

Thanks to the reviewer for this question. According to question 8, we have added the information about the sensing materials and the sensing principle of gas sensors in the revised manuscript.

In this article, our intention is to differentiate the odors of human being and other common objects in order to achieve object recognition in the burial scenarios. Therefore, we chose the mentioned 11 objects, including orange, towel, stone, can, worn clothes, carton, mug, mouse, hair, leg and arm in the experiment of this article. The results show that the six-channels gas sensing arrays we used could efficiently accomplish the goal.

Several works have shown that ammonia is one of the important components of human body's odor.

DOI: 10.1021/acs.est.0c00094.

DOI: 10.1038/s41598-020-68860-z.

DOI: 10.1016/j.jchromb.2017.03.034.

Therefore, we chose ammonia as the interference gas to test the anti-interference ability of our system for object recognition in Figure 4e.

We have added corresponding discussion in the revised manuscript (Please see Page 6).

10) “the force sensor can distinguish objects with various stiffness, thus distinguishing the soft human arm from the rigid debris” is not the reason for the tactile olfactory fusion method maintains its initial high accuracy (Figure 4e)

Response:

Thanks to the reviewer for this comment. This comment came from an unclear statement in our previous manuscript and we are sorry for the confusion caused by the lack of detailed description. “the force sensor can distinguish objects with various stiffness, thus distinguishing the soft human arm from the rigid debris” is the reason for high accuracy of tactile perception only when there is no block for the mechanical hand.

“by increasing the weight proportion of olfactory information, identification of human odor can be a strong supplement to the lack of tactile perception” is the reason for the tactile olfactory fusion method maintains its initial high accuracy and lead to an improvement in the accuracy of human body recognition in burial scenarios.

We have changed the relevant expressions in the revised manuscript to avoid any misleading (Please see Page 10).

11) It is mentioned “improve accuracy via changing the tactile and olfactory weight”. Few words should be added to describe the principle and significance of this operation.

Response:

We sincerely thank the reviewer for the insightful suggestion. The operation of improving accuracy via changing the tactile and olfactory weight is mainly in the multimodal fusion algorithm, as shown in equation (1), which is defined as

$$I_{fusion} = MCB(T_{net}, O_{net}, n_t, n_o, d) \quad (1)$$

where T_{net} and O_{net} are the tactile and olfactory features extracted from CNN, n_t is the length of tactile feature vector, n_o is the length of olfactory feature vector and d is the length of fusion feature vector. All of them are environmental parameters obtained from the supervised scenario-dependent feedback. We can rewrite above equation specifically as

$$I_{fusion} = FFT^{-1} \left(FFT \left(\Psi(Resize(T_{net}, n_t)) \right) \odot FFT \left(\Psi(Resize(O_{net}, n_o)) \right) \right) \quad (2)$$

where FFT means fast Fourier transform, and $length(I_{fusion}) = length(\Psi) = d$, the *Resize* function adjusts the feature vector according to the proportionality coefficients of tactile and olfactory vectors in the fusion process. This operation alleviates the impact caused by

interference and increases the data representativeness.

We have added corresponding discussion in the revised manuscript (Please see Page 18 and 19).

12) In Figure 4i, the rescue of surviving human beings is not secure, because in reality, people are not only partially buried. It is recommended to delete this figure.

Response:

Thanks to the reviewer for this comment. This protocol is not a complete rescue plan which could be directly used for human research and rescue in a complex scene of a real disaster. The rescue procedures shown in Figure 4i is only a corresponding protocol for the specific situation in our experiments, which follows the guidelines of our local fire department; such procedures aim to reflect the capability of our system as assisting human rescue in burial scenarios.

We have changed the relevant expressions in the revised manuscript to avoid any misleading (Please see Figure 4i).

13) The resulting outcome is very significant for the development of robust object recognition in harsh environments. Unfortunately, the writing ability of this manuscript is approximately terrible. As a scientific article, the main text should focus on the in-depth mechanism instead of experimental phenomenon description. For example, the force sensing mechanism, as well as the realization method of the susceptible gas sensor with specificity in this work, should be detailedly discussed. Worst of all, the result of the BOT-M with highest recognition rate is also with no interpretation. Here, I strongly recommend adding these discussion.

Response:

We sincerely thank the reviewer for the insightful suggestion. We have added supplementary explanation about these points the reviewer mentioned above in order to improve the quality of our manuscript.

Here, **Figure R9** shows the sensitive hexagonal silicon membrane within the force sensor, on which located a Wheatstone bridge made by four piezoresistors. In principle, when the force sensor is in contact with the detected object, the membrane would form a certain deformation, the degree of which depends on both the stiffness of the object and the force applied. Then the membrane deformation would cause the changes in resistance values of the four piezoresistors. Thus, the voltage output value of the Wheatstone bridge could reflect the change of resistance value and the degree of membrane deformation. Based on this working principle, **when we use the mechanical hand to touch the object, the output voltage value distinguishes the local topography and stiffness of various objects**, providing important information for the following object recognition process.

We have added corresponding discussion in the revised manuscript (Please see Page 6 and 35-37, Note 2 and 3, Supplementary Figure 3, Supporting Information).

Figure R9. Interior structure of a single force sensor. a) SEM image showing the sensitive hexagonal silicon membrane within the force sensor, on which located a Wheatstone bridge made by four piezoresistors. b) Schematic illustration of the electronic circuit of the four piezoresistors.

Additionally, we have added interpretations for the result of the BOT-M with highest recognition rate. Firstly, we process input information with CNN, which is proved to be effective in other multimodal object recognition models, to extract the key feature vectors for subsequent processing. Secondly, we present a multimodal fusion algorithm based on MCB, which can deeply fuse the tactile and olfactory feature vectors, instead of concatenation, element-wise product, element-wise sum or other simple arithmetic operations. This is a good method to optimize the data representation and improve the model performance. Thirdly, the multiple classification algorithm is proposed to calculate the recognition results by sampling several times to avoid accidental errors and improve the accuracy. Finally, we can adjust the proportion of tactile and olfactory information through the supervised scenario-dependent feedback to adapt to different actual applications and provide reference for final decision-making.

14) What is the unique gas of human beings? Could this design discriminate living person from dead bodies? Animal should be added as interference objects because it is the most human-like one.

Response:

Thanks to the reviewer for this question. Similar to comment 9, several works have shown that ammonia and hydrogen sulfide are two of the important components of human body odor.

DOI: 10.1021/acs.est.0c00094.

DOI: 10.1038/s41598-020-68860-z

DOI: 10.1016/j.jchromb.2017.03.034

Besides, some studies have shown the odor of dead body.

DOI: 10.25148/etd.FI14040888

<https://digitalcommons.fiu.edu/cgi/viewcontent.cgi?article=2327&context=etd>

Due to experimental conditions and human ethics restrictions, we couldn't do the

experiments to obtain human dead body's odor. However, in principle, because the odor composition and concentration of the dead body is different from that of living person, it is very likely that our gas sensors could distinguish the difference between these odors based on the working principle mentioned before in comment 8.

15) The following related references, which have been published recently, could be cited to highlight the significance of developing multimode sensing array with bionic strategy (Advanced Materials, 2020, 32, 1907043).

Response:

We sincerely thank the reviewer for providing us with these high-quality references. We have thoroughly read them and found them influential. We have added them in our reference list in order to highlight the significance of developing multimode sensing array with bionic strategy.

16) In order to consolidate the excellent property of robotic arm and make the main purpose of this article unambiguous, the video related with Figure 4h should be added as supporting file.

Response:

We sincerely thank the reviewer for this insightful suggestion. In order to show the autonomy level of the robotic motion, we have recorded a video about various tasks mentioned in Fig 4h, including Removing debris and Human detection by the mechanical hand. As shown in the video, we mounted the mechanical hand on a robotic arm to recognize and grab the debris, and then move them away from the designated area; then we let the same robotic system to touch and recognize the fully-exposed human arm/leg.

We have added corresponding discussion in the revised manuscript (SI video 1 and 2).

Reviewers' Comments:

Reviewer #1:

Remarks to the Author:

In the first round of reviews of this manuscript, the key questions centered on two themes. First, given that multimodal sensing has been previously used extensively, the authors were asked to demonstrate the key advantages of the combination of tactile sensing & olfactory sensors (especially considering other recent multimodal sensing in literature e.g., G. Li et al. *Sci Robotics* 2020). These queries were to ascertain the novelty of this work over other current literature and the benefits of olfaction in comparison to other modalities. Second, my review (and others) probed the rigors of the authors technical claims: including the ability to sense texture with 14 tactile sensors, or the basis for the choice of gas sensors etc. and its inner workings.

I want to thank the authors for their attempts to address these questions. The original work was reasonably well-presented, and I felt the authors deserved an opportunity to address the major criticisms. However, in my opinion this revision does not add to the technical rigor or fully address the key lingering concerns that were raised in the first round. The benefits of the combining 14 tactile sensors/finger with 6 gas sensors is not clear still. How the specific gas sensors used here help in resolving different objects is not clear. Moreover, I find that several claims around ability to sense topography/texture with 14 tactile sensors alone are not completely justified. Furthermore, the level of presentation of new experimental results only raises more questions. For example, simply looking at the new data in showing ability to sense topography: (a) why should a row of 14 sensors respond differently when the same force is applied across different step heights? In all these cases, the same number of sensors appear to support the load. More fundamentally, are the sensors picking up the force, or deflection or something else? (b) What is the ground truth when showing "texture" measurements of apple vs. orange and why do all the 5 sensors show the same data with the orange are the five fingers independently contacting the same area?

Overall, I think the combination of tactile sensing and olfaction may prove to be interesting in the future which was part of my initial consideration. But I'm not convinced this specific demonstration meets the rigors of characterization, presentation and validation usually seen in this journal.

Reviewer #2:

Remarks to the Author:

The authors well addressed my previous concerns that I would like to recommend it to be accepted by Nature Communications.

Reviewer #3:

Remarks to the Author:

I have reviewed the revision of this manuscript. The authors have answered and modified my comments accordingly, the text and figures are re-organized properly. Up to now, I think this work has met the requirements of scientific level, novelty and quality of Nature Communications. Hence, I suggest it is publishable without further revision.

Point by point response (comments in black and responses are in blue):

Reviewer #1:

In the first round of reviews of this manuscript, the key questions centered on two themes. First, given that multimodal sensing has been previously used extensively, the authors were asked to demonstrate the key advantages of the combination of tactile sensing & olfactory sensors (especially considering other recent multimodal sensing in literature e.g., G. Li et al. Sci Robotics 2020). These queries were to ascertain the novelty of this work over other current literature and the benefits of olfaction in comparison to other modalities. Second, my review (and others) probed the rigors of the author's technical claims: including the ability to sense texture with 14 tactile sensors, or the basis for the choice of gas sensors etc. and its inner workings.

I want to thank the authors for their attempts to address these questions. The original work was reasonably well-presented, and I felt the authors deserved an opportunity to address the major criticisms. However, in my opinion this revision does not add to the technical rigor or fully address the key lingering concerns that were raised in the first round.

Response:

We would like to express our sincere thanks to the reviewer for carefully considering our revised manuscript in the first round and encouraging that the original work was reasonably well-presented.

Each of the following comments is highly important to further improve the quality of our manuscript. The followings are the point-to-point replies to major concerns (**divided into sentences**) from the comments.

The benefits of the combining 14 tactile sensors/finger with 6 gas sensors is not clear still.

Response:

Thanks to the reviewer for this comment. We have added more detailed explanation focused on the benefits of this combination of tactile sensor array and gas sensor array in order to further highlight the advantages of the system reported in this work.

For the purpose of object recognition, this combination strategy can **carry more useful information** of the detected objects and help achieve more accurate identification results **compared to other physical parameters, like temperature and humidity. Odor is the chemical fingerprint of every object.** Thus, compared to other ordinary physical parameters (e.g. temperature or humidity, e.g., G. Li et al. Sci Robotics 2020), **the olfactory sensing detects 'orthogonal' features of the object and could differentiate tactilely similar objects with different compositions, such as mice and human.** Hence, in principle, the combination strategy of tactile and olfactory sensing achieves higher accurate identification and anti-interference ability (object recognition with 96.9%

accuracy for the test dataset and with > 80% accuracy in rescuing scenarios) compared to other combination strategies. See Figure 3c and 4e in the manuscript.

In tactile perception, we mainly focused on the detection of the **local topography (roughness) and material stiffness of the detected objects in order to distinguish human from other objects**. Compared to the traditional shape-based objects recognition through tactile perception (e.g., Wang, M. et al. Nat. Electron 2020), our method requires a smaller tactile sensing array. Moreover, in the design of the distribution of tactile sensors, we located the sensing arrays on the fingertips of the mechanical hand to gain information of local topography (roughness) and material stiffness, mimicking the tactile perception of the star-nosed mole (Nat. Neurosci. 4, 353-354, 2001). This design facilitates objects recognition solely by touching the objects without the needs of actual grabbing, resulting in a more convenient manipulation and data processing. **According to the limited area of each fingertip, 14 force sensors have been chosen in our design as a trade-off to form a single sensing array, which can acquire sufficient information for further objects recognition.**

In olfactory perception, six gas sensors have been chosen for the detection of six specific gases, including ethanol, acetone, ammonia, carbon monoxide, hydrogen sulfide, and methane, which play an important role in object recognition. For example, ammonia and hydrogen sulfide are common components of biological odors, which could help to distinguish animal and human from other objects (See Figure R1). Furthermore, by differentiating the concentration of these two gases, human can be distinguished from other animals such as mice (See confusion matrix from Figure 3c in the manuscript, page 31). Thus, based on this six-channel olfactory sensing array, efficient odor recognition of different objects can be achieved.

Figure R1. The hexagonal olfactory mappings of three different objects including an arm, worn clothes, and an orange. Please see Figure 2h in manuscript.

Therefore, this combination strategy (14 tactile sensors on each finger and 6 gas sensors) could help to reduce the input data size ($5 \times 14 + 6$) in the process of objects recognition, leading to a contracted consumption of computing resource and fast

identifications (within one second). Vision-based sensing systems have been widely used for object recognition purpose, which could achieve accurate identification by processing the captured images. However, image processing always requires a large input data size and computing resources. In comparison, the faster identification of tactile and olfactory sensing combination strategy presents great superiority especially in the rescue mission, which requires fast decision making in low-visibility environments. See Figure 4 in the manuscript.

Furthermore, this tactile-olfactory combination strategy could also help to achieve anti-interference ability in challenging scenarios and could offer excellent human identification performance (accuracy > 80%) under the hazardous scenarios of gas interference, object burying, and damaged sensors. According to actual applications, when one input perception is severely disturbed or damaged by the environment, this system still allows compensation from the other perception and ensures a high recognition rate for objects. For example, the interference of olfactory perception can be alleviated and a relatively high recognition accuracy can be maintained when the target object is covered with water or muddy water. See Figure R2.

Figure R2. The recognition accuracy of orange covered with water and muddy water using olfactory-based recognition and BOT-M associated learning. Please see Supplementary Figure 11 in manuscript

We have added corresponding discussion in the revised manuscript (Please see Page 37 and 39, Supporting Information).

How the specific gas sensors used here help in resolving different objects is not clear.

Response:

Thanks to the reviewer for this comment. Here we have added supplementary information in order to better explain the functions of specific gas sensors for identifying different objects.

Notably, we chose a specific combination of 6 gas sensors in order to accurately recognize and differentiate human from other common interferences in rescue scenarios. Those six gas sensors have been chosen for the detection of six specific gases, including ethanol, acetone, ammonia, carbon monoxide, hydrogen sulfide, and methane, which play an important role in object recognition. For example, ammonia and hydrogen sulfide are two common components in biological odors (DOI:10.1021/acs.est.0c00094; DOI: 10.1038/s41598-020-68860-z), which could help to distinguish animal and human from other objects. Also, by differentiating the concentration of these two gases, human can be distinguished from other animals such as mice according to the results in this article. In addition, the other four gases, including ethanol, acetone, carbon monoxide, and methane, are common dangerous gases appeared in rescue scenarios; the danger level of the surrounding environment can be identified by detecting and recognizing these gases. Therefore, this specific gas sensors combination design can successfully achieve the goal of human identification and rescue in dangerous scenarios.

Based on the above-mentioned principle, specific six sensing materials have been chosen for our olfactory sensing array: (1) carbon nanotubes modified by magnesium oxide particles; (2) carbon nanotubes modified with platinum particles; (3) graphene modified with copper oxide particles; (4) platinum-doped tin oxide; (5) platinum-doped tungsten oxide; (6) composite material of zinc oxide and tin oxide. Based on the gas sensor working principle mentioned in the manuscript as well as the former response letter in the first round, **since the sensitivities of each material to different gas molecules are different, this complementary setup (6 gas sensors) could help to sense the type and concentration of various gases, reflecting the odor profile of the detected object.**

Besides, based on the above-mentioned working principle of our six-channel gas sensor array, by identifying the odors of different objects, **the olfactory perception could help to resolve recognition between objects with similar tactile characteristics.** For example, since the stiffnesses of human and animal body can be very similar, it's challenging to identify them by tactile perception alone. In this situation, these specific gas sensing arrays could help to distinguish human and animal body only through odor recognition. Also, according to actual applications, when tactile perception is severely disturbed or damaged by the environment, this system still allows compensation by olfactory perception and ensures a high recognition accuracy.

Moreover, I find that several claims around ability to sense topography/ texture with 14 tactile sensors alone are not completely justified. Furthermore, the level of presentation of new

experimental results only raises more questions. For example, simply looking at the new data in showing ability to sense topography:

(a) why should a row of 14 sensors respond differently when the same force is applied across different step heights? In all these cases, the same number of sensors appear to support the load.

Response:

Thanks to the reviewer for this question. This question may result from the unclearness of the former schematic illustration and plotting. Here we have adjusted the illustration and added some more detailed explanation about the specific force analysis of tactile sensor arrays when touching different objects.

First, as shown in Figure R3 (a), the thickness of each force sensor is around 0.1 mm and every sensor is covered by soft silica gel to prevent them from damages caused in the touching process as mentioned in the experimental section. Thus, when one tactile sensor array (a row of 14 force sensors) is in contact with the surface of the detected object, **not only the 14 sensors are in contact with the object, but the substrate (blue layer in Figure R3 (a)) will also be in contact with the object at the same time. In this case, different contact surfaces will lead to different force distribution on the force sensors and the substrate below. Therefore, the 14 force sensors respond differently when the same overall force is applied across different step heights—we can only quantitatively apply an overall force on the whole sensor array—because of the variation of force dispersion on the substrate.**

Moreover, as shown in Figure R3 (b), F_{active} represents the average applied force and F_{reactive} **demonstrates the reactive force from the object to one single sensor.** When the same force is applied to different objects by the mechanical hand, the deformation degree of the objects varies according to objects' different elasticity moduli, which would change the local contact area and result in a different value of F_{reactive} . For example, when contacting an ideally rigid object with zero deformation, F_{active} is the same as F_{reactive} ; when contacting a soft object with large deformation, the overall reactive force disperses both on the sensor (F_{reactive}) and the substrate, resulting in $F_{\text{active}} > F_{\text{reactive}}$. Similarly, different step heights (shown in Figure R3 (c)) could also change the local contact area, leading to different results of sensor response (F_{reactive}). **Thus, in practical situations, different loads are supported by different sensors that reflect the local topography and material stiffness of the touched object.**

We have added corresponding discussion in the revised manuscript (Please see Page 36 and 37, Supporting Information).

Figure R3. Tactile perception of different materials and height gradient. a) Schematic illustration of using a single tactile sensing array (1*14) to detect height gradient. b) Response curve of the force sensor during the process of contacting three different objects. When the same force is applied to the objects by the mechanical hand, the deformation degree of the objects varies according to their different elasticity modulus, changing the local contact area and consequent reactive pressure. c) Output voltage of force sensors while touching paper pile with thickness from 0.2mm to 1.5mm.

More fundamentally, are the sensors picking up the force, or deflection or something else?

Response:

Thanks to the reviewer for this question. **The force sensor only picks up the reactive force applied upon the sensor.** As shown in Figure R4 (a) and (b), when a certain force is applied upon the force sensor, the sensitive hexagonal silicon membrane (Figure R4 (a)) would form a certain deformation (not the deformation of the object), which would cause the changes in resistance values of the four piezoresistors. In this situation, the voltage output value of the Wheatstone bridge could reflect the degree of membrane deformation which depends on the value of the applied force. Nevertheless, the picked force information reflects the information of local topography (roughness) and material stiffness for the detected objects.

Figure R4. Interior structure of a single force sensor. a) SEM image showing the sensitive hexagonal silicon membrane within the force sensor, on which located a Wheatstone bridge made by four piezoresistors. b) Schematic illustration of the electronic circuit of the four piezoresistors.

(b) What is the ground truth when showing “texture” measurements of apple vs. orange and why do all the 5 sensors show the same data with the orange are the five fingers independently contacting the same area?

Response:

Thanks to the reviewer for this question. This confusion might be because of the lack of description of both the touching process and the tactile mapping results. We have added more detailed information in order to better explain the “texture” measurements procedure.

During the experiment, we used the mechanical hand to grab the detected objects, which are apple and orange in this case, thus the five fingertips contacted with different areas of the objects’ surface simultaneously. As shown in tactile mappings from Figure R5 (a) and (d), there are five rows (five finger-tips) in each mapping and each row contains 14 squares (14 force sensors on each finger-tip), the color of which reflect the force values applied on each sensor from the tactile sensing array on one finger-tip. The results from Figure R5 (b) and (e) present the force values of 14 sensors from the sensing array (a row of 14 sensors) on finger-tip #1. Similar curves can be plotted for the other four finger-tips based on the tactile mapping results. For example, Figure R5 (c) and (f) present the force values of 14 sensors from the sensing array on finger-tip #3, which is different from finger-tip #1.

In this case, tactile sensing arrays can accurately present the surface roughness differentiation between the tested objects (Figure R5 (b) and (e)). Therefore, the local topography of tested objects can be obtained from the five tactile sensing arrays mounted at each finger-tips of a mechanical hand (Figure R5 (a) and (d)).

Figure R5. Tactile perception of different objects with various surface roughness. a) The tactile mapping of the local topography of an apple surface, detected by the tactile sensing arrays (5*14) mounted on the fingertips of a mechanical hand. b, c) Output voltage of every force sensor from a single tactile sensing array (1*14), which presents the local roughness of the apple surface. d) The tactile mapping of the local topography of an orange surface, detected by the same tactile sensing arrays (5*14). e, f) Output voltage of every force sensor from a single tactile sensing array (1*14), which presents the local roughness of the orange surface.

Overall, I think the combination of tactile sensing and olfaction may prove to be interesting in the future which was part of my initial consideration. But I'm not convinced this specific demonstration meets the rigors of characterization, presentation and validation usually seen in this journal.

Response:

The main innovations of our work are summarized as follows (See Figure R6):

1) Tactile-olfactory bionic sensing. We have proved that the **olfactory sensing**, which has not been combined with tactile sensing for object identification before, can also be **suitable for conducting features fusion** with tactile data that achieve **object recognition with 96.9% accuracy for the test dataset and with > 80% accuracy in rescuing scenarios.**

2) High-performance sensing array design. In our design, we use silicon-based force sensors fabricated by micro-electromechanical systems (MEMS) technologies, which have more robust performance and finer spatial resolution compared to other flexible devices (e.g., G. Li et al. Sci Robotics 2020), and can provide information of **local**

topography (roughness) and material stiffness. Please see corresponding responses for updated discussion.

3) **Bioinspired olfactory-tactile associated machine-learning algorithm.** Compared with the vision-based sensing systems, our tactile and olfactory fusion strategy has a relatively **small input data size (5×14+6)**, leading to a **contracted consumption of computing resource and fast identifications (within one second)**, which are crucial in proceeding rescue missions. See SI video 1 and 2.

4) **Achieving human identification during rescue missions in challenging environments.** While combining olfactory sensing with tactile perception, our sensing system also demonstrates the **anti-interference ability** in challenging scenarios and could offer excellent human identification performance (accuracy > 80%) under the hazardous scenarios of gas interferences, debris burying, damaged sensors, and rescue mission. Thus, our work is **application-oriented with a specific focus on supporting human recognition and rescue in dangerous scenarios.**

Figure R6. The summary of main innovations of our work. Proposed framework addressing the main innovations and major features of tactile-olfactory bionic sensing.

In general, we highly appreciate the great efforts that the reviewer has devoted to our manuscript and the essential interests and importance that the reviewer has acknowledged towards our work. We have added more detailed explanation in order to improve our manuscript and hopefully to meet the rigors of characterization, presentation and validation of **Nature Communications**. Please check page 36-39.

Reviewer #2 (Remarks to the Author):

The authors well addressed my previous concerns that I would like to recommend it to be accepted by Nature Communications.

Response:

We would like to express our sincere thanks to the reviewer for the positive evaluation of our work.

Reviewer #3 (Remarks to the Author):

I have reviewed the revision of this manuscript. The authors have answered and modified my comments accordingly, the text and figures are re-organized properly. Up to now, I think this work has met the requirements of scientific level, novelty and quality of Nature Communications. Hence, I suggest it is publishable without further revision.

Response:

We would like to express our sincere thanks to the reviewer for carefully considering our revised manuscript in the first round and positively evaluating that the text and figures are re-organized properly.

Reviewers' Comments:

Reviewer #1:

Remarks to the Author:

I have no additional comments.